# Ubiquitination of RIPK1 regulates its activation mediated by TNFR1 and TLRs signaling in distinct manners

Xingyan Li[1], Mengmeng Zhang[1], Xinyue Huang[1], Wei Liang[1], Ganquan Li[1], Xiaojuan Lu[1], Yanxia Li[1], Heling Pan[1], Linyu Shi[1], Hong Zhu[2], Lihui Qian[1], Bing Shan[1✉] & Junying Yuan[1✉]

RIPK1 is a death-domain (DD) containing kinase involved in regulating apoptosis, necroptosis and inflammation. RIPK1 activation is known to be regulated by its DD-mediated interaction and ubiquitination, though underlying mechanisms remain incompletely understood. Here we show that K627 in human RIPK1-DD and its equivalent K612 in murine RIPK1-DD is a key ubiquitination site that regulates the overall ubiquitination pattern of RIPK1 and its DD-mediated interactions with other DD-containing proteins. K627R/K612R mutation inhibits the activation of RIPK1 and blocks both apoptosis and necroptosis mediated by TNFR1 signaling. However, *Ripk1*$^{K612R/K612R}$ mutation sensitizes cells to necroptosis and caspase-1 activation in response to TLRs signaling. *Ripk1*$^{K612R/K612R}$ mice are viable, but develop age-dependent reduction of RIPK1 expression, spontaneous intestinal inflammation and splenomegaly, which can be rescued by antibiotic treatment and partially by *Ripk3* deficiency. Furthermore, we show that the interaction of RIPK1 with FADD contributes to suppressing the activation of RIPK3 mediated by TLRs signaling. Our study demonstrates the distinct roles of K612 ubiquitination in mRIPK1/K627 ubiquitination in hRIPK1 in regulating its pro-death kinase activity in response to TNFα and pro-survival activity in response to TLRs signaling.

[1] Interdisciplinary Research Center on Biology and Chemistry, Shanghai Institute of Organic Chemistry, Chinese Academy of Sciences, 26 Qiuyue Rd, PuDong District, 201203 Shanghai, China. [2] Department of Cell Biology, Harvard Medical School, 240 Longwood Ave., Boston, MA 02115, USA. ✉email: shanbing@sioc.ac.cn; junying_yuan@sioc.ac.cn

Receptor interacting protein kinase 1 (RIPK1) is an important mediator of multiple signaling pathways activated by the members of death receptor family and pattern recognition receptors. RIPK1 contains a N-terminal kinase domain, a C-terminal death-domain (DD) and an intermediate domain that contains a RIP homotypic interaction motif (RHIM). In TNFα-stimulated cells, RIPK1 kinase performs pro-cell-death and pro-inflammatory activities by activating RIPK1-dependent apoptosis and necroptosis[1,2]. However, the scaffold of RIPK1 is also known to perform pro-survival function independent of its kinase activity. RIPK1 deficiency in rare human cases leads to immunodeficiency and inflammatory bowel disease (IBD)[3–5]. $Ripk1^{-/-}$ mice die at birth with systemic inflammation, which can be reduced by knockout of Myd88 or TRIF, the key mediators of TLR3/4 signaling[6,7]. The perinatal lethality of $Ripk1^{-/-}$ mice and increased cellular sensitivity to necroptosis of $Ripk1^{-/-}$ cells can be blocked by double deficiency of $Ripk3$ and $casp8$[8]. Thus, the kinase activity of RIPK1 promotes necroptosis while the scaffold function of RIPK1 suppresses necroptosis. It is unclear, however, how RIPK1 is regulated to serve these two distinct functions.

DD is a key protein–protein interaction module that mediates both homotypic as well as heterotypic interactions between different DD-containing proteins in TNFα and TLRs signaling pathways. The DD at the C-terminus of RIPK1 mediates its recruitment into TNFR1 complex and also for the activation of RIPK1 kinase by dimerization[9,10]. In TNFα stimulated cells, RIPK1 and TRADD, an adapter protein with a DD, bind with the DD in the intracellular domain of TNFR1 to form a key signaling complex (TNF-RSC/complex I)[11]. In complex I, RIPK1 is extensively modified by different types of ubiquitin chains, including M1, K11, and K63, mediated by E3 ubiquitin ligases, such as cIAP1/2 and linear ubiquitin chain assembly complex (LUBAC)[12]. The ubiquitin modifications on RIPK1 serve as anchors to recruit different ubiquitin binding proteins such as TAB2/3 and NEMO which in turn promote the activation of TAK1 and IKK complex, respectively, to restrict the activation of RIPK1 kinase and promote NF-κB signaling[13–19]. When the ubiquitination of RIPK1 in complex I is reduced by elimination of cIAP1/2 or LUBAC, RIPK1 kinase is activated to form complex IIa with caspase-8 and FADD, another adapter protein with a DD, to mediate RIPK1 kinase dependent apoptosis (RDA)[19–22]. FADD deficiency can promote the formation of RIPK1/RIPK3/ MLKL complex, termed as complex IIb or necrosome, to mediate necroptosis[23–27]. Thus, the activation of RIPK1 is regulated by complex ubiquitination modifications, and the interaction with multiple DD-containing proteins, including TRADD, FADD, and TNFR1 as well as itself. However, it is still not clear how RIPK1-DD interacts with other DD-containing proteins and the mechanisms of ubiquitination that regulate the activation of RIPK1 kinase upon different cellular stimulations.

Stimulation of TLR4 by LPS promotes the activation of caspase-1 mediated by NLRP3 inflammasome[28]. Activated caspase-1 in turn cleaves pro-IL-1β to generate mature IL-1β to mediate inflammatory response. Activation of RIPK3 has been shown to contribute to the inflammasome pathway in bone marrow-derived dendritic cells (BMDCs) as RIPK3 knockout can inhibit LPS-induced IL-1β secretion[29]. LPS stimulation promotes the formation of a signaling complex involving RIPK1, RIPK3, FADD, and caspase-8. In this complex, the scaffold of RIPK1 negatively regulates the activation of RIPK3. However, it remains unclear how the scaffold function of RIPK1 suppresses the activation of RIPK3 to regulate caspase-1 activation and IL-1β cleavage.

Here, we conducted a systematic genetic mutagenesis study to determine the functional significance of different ubiquitination sites on RIPK1. We identify that K627 in human RIPK1 and its conserved residue K612 in murine RIPK1 in the DD are important for mediating RIPK1 ubiquitination, homodimerization as well as heterodimerization of RIPK1 with TNFR1, TRADD, and FADD. We show that K612 in mRIPK1/K627 in hRIPK1 is a key ubiquitination site that regulates the overall ubiquitination pattern of RIPK1. By establishing and analyzing K612R knockin mice, we show that K612R mutation inhibits RIPK1 activation and thus blocks RDA and necroptosis in response to TNFα stimulation. Surprisingly, $Ripk1^{K612R/K612R}$ bone marrow-derived macrophages (BMDMs) are hypersensitive to LPS or Poly (I:C) induced necroptosis and caspase-1 activation with increased RIPK3/RIPK1/MLKL necrosome formation. Although $Ripk1^{K612R/1K612R}$ mice are born normal, they develop age-dependent reduction of RIPK1 expression, intestinal inflammation, and splenomegaly characterized by macrophage infiltration and excessive cytokine production, which can be rescued by antibiotic treatment and ameliorated by RIPK3 deficiency. We show that the failure of K612R RIPK1 to interact with FADD serves a critical role in sensitization of $Ripk1^{K612R/K612R}$ mutant cells to necroptosis and caspase-1 activation induced by TLR3/4. Thus, ubiquitination of K612 in mRIPK1/K627 in hRIPK1 serves distinct regulatory functions in response to TNFR1 and TLRs signaling.

## Results

**K612 in mRIPK1/K627 in hRIPK1 is important for mediating RDA and necroptosis induced by TNFα.** Previous proteomic studies have shown that RIPK1 is extensively ubiquitinated at multiple sites including K115, K307, K377, K316, and K627[30–33], although the functional significances of these ubiquitination events, other than that of K377 which is involved in mediating NF-κB activation[33], are largely unclear. To explore the functional significance of different ubiquitination events, we reconstituted $Ripk1^{-/-}$ MEFs with WT RIPK1 or RIPK1 mutants with individual ubiquitination site mutated to Arg (K to R), including K20R, K105R, K392/K395R, and K612R. Among the ubiquitinating site mutants screened, we found that only RIPK1 K612R complemented MEFs showed significant resistance to both RDA induced by TNFα/SM-164 or TNFα/5Z-7 and necroptosis induced by the additional presence of pan caspase inhibitor zVAD.fmk (Supplementary Fig. 1a). The protein expression levels of RIPK1 mutants in complemented $Ripk1^{-/-}$ MEFs were comparable with that of WT RIPK1 shown by western blotting (Supplementary Fig. 1b). K612 in murine RIPK1 and its equivalent residue K627 in human RIPK1 are localized in the DD. To examine the role of K627 in human RIPK1, we reconstituted Jurkat Ripk1 deficient cells with WT or K627R RIPK1 and found that K627R RIPK1 reconstituted Jurkat cells were also resistant to necroptosis induced by TNFα/SM-164/zVAD.fmk or TNFα/5Z-7/ zVAD.fmk (Supplementary Fig. 1c, d).

**RIPK1$^{K612R}$ mutant cells are resistant to necroptosis induced by TNFα.** To characterize the mechanism by which RIPK1$^{K612R}$ mutation affects the activation of RIPK1, we generated $Ripk1^{K612R/K612R}$ knockin mutant mice by mutating K612 to Arg using CRISPR/Cas9 technology (Supplementary Fig. 2). We examined the effect of K612R mutation in immortalized MEFs generated from littermates of $Ripk1^{+/K612R}$ parents, primary BMDMs to different pro-necroptosis and pro-RDA stimuli, including TNFα/SM-164/zVAD.fmk, TNFα/5Z-7/zVAD.fmk, TNFα/CHX/zVAD.fmk, SM-164/zVAD, 5Z-7/zVAD, TNFα/SM-164, and TNFα/5Z-7. $Ripk1^{K612R/K612R}$ MEFs and primary $Ripk1^{K612R/K612R}$ BMDMs showed significant resistance to necroptosis and RDA triggered by TNFα in combination (Fig. 1a–c). In contrast, $Ripk1^{K612R/K612R}$ MEFs did not show any

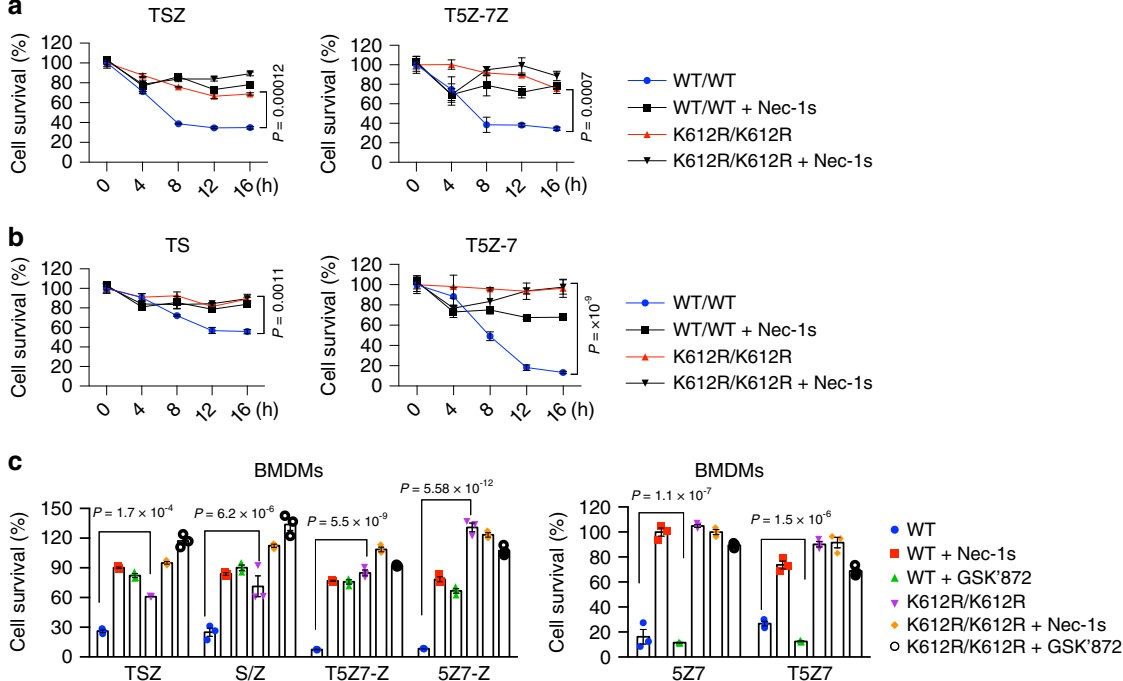

**Fig. 1 RIPK1[K612R] mutant cells are resistant to necroptosis induced by TNFα. a**, **b** $Ripk1^{+/+}$ and $Ripk1^{K612R/K612R}$ immortalized MEFs were pretreated with SM-164 (100 nM) or 5Z-7 (5Z 7-oxozeanol) (200 nM) for 1.5 h in the presence or absence of Nec-1s (10 μM) as indicated, and then treated with TNFα (100 ng/ml)/Z-VAD (25 μM) (**a**) or TNFα (100 ng/ml) (**b**) for different time points. Cell survival was measured by Cell TiterGlo. Data are presented as mean ± SEM of n = 3 biologically independent samples. Two-way ANOVA with Bonferroni's multiple comparison test. **c** WT and RIPK1-K612R primary BMDMs were pretreated with SM-164 (100 nM) or 5Z-7 (200 nM) for 1.5 h in the presence or absence of Nec-1s (10 μM) or GSK'872 (10 μM) as indicated, and then treated with TNFα (100 ng/ml) alone or Z-VAD (25 μM) alone or Z-VAD (25 μM) plus TNFα (100 ng/ml) for 10 h. Cell survival was measured by Cell TiterGlo. Data are presented as mean ± SEM of $n = 3$ biologically independent samples. Two-way ANOVA with Bonferroni's multiple comparison test.

resistance to RIPK1-independent apoptosis induced by TNFα/CHX (Supplementary Fig. 3a), but showed some protection against TNFα/CHX/Z-VAD induced necroptosis (Supplementary Fig. 3b). Taken together, these results suggest that K627 and K612 in DDs of human RIPK1 and murine RIPK1, respectively, are important in mediating RDA and necroptosis in response to TNFR1 signaling.

**K612R mutation in mRIPK1 alters complex I and disrupts complex II formation downstream of TNFR1 activation by TNFα.** We investigated the mechanism by which K612R mutation blocked RDA and necroptosis. In response to pro-RDA stimuli, activation of RIPK1 promotes the formation of complex IIa, which contains activated RIPK1, FADD, and caspase-8, to mediate the activation of caspase-8 and apoptosis. We found that the activation of RIPK1, as shown by its activation biomarker p-S166[34], in cells stimulated by T/S and T/5Z-7 was inhibited by K612R mutation (Fig. 2a, b). The cleavage of caspase-8, caspase-3, and RIPK1 was also inhibited by RIPK1 kinase inhibitor Nec-1s and K612R mutation. Consistent with inhibition of RIPK1 activation and caspase-8 cleavage, the formation of complex IIa, as shown by the interaction of FADD with RIPK1 and caspase-8, in response to T/S or T/5Z-7 stimulation, was also inhibited by K612R mutation (Fig. 2a, b).

Under apoptosis deficient conditions, activated RIPK1 interacts with RIPK3 to promote its activation which consequently mediates the binding and phosphorylation of MLKL and subsequent formation of Complex IIb, which is critical for the execution of necroptosis[35,36]. The activation of RIPK1 as well as downstream phosphorylation events including p-T231/S232

RIPK3 and p-S345 MLKL in cells stimulated by pro-necroptotic T/S/Z and T/5Z-7/Z conditions were reduced by K612R mutation (Fig. 2c, d). The formation of complex IIb, as shown by the binding of RIPK3 with RIPK1 and MLKL, was inhibited by K612R mutation (Fig. 2c, d).

In TNFα stimulated cells, RIPK1 and a DD-containing adapter protein TRADD are recruited to interact with the DD of TNFR1 to form a transient TNFR1 signaling complex (TNF-RSC/complex I) along with other components including cIAP1/2, A20, and linear ubiquitin chain assembly complex (LUBAC)[11]. We next investigated the effect of RIPK1 K612R mutation on the recruitment of RIPK1 and other components into complex I. The complex I was immunoprecipitated by TNFR1 antibody in MEFs stimulated by TNFα or by Flag agarose beads in MEFs stimulated by Flag-TNFα alone or in combination with SM-164, or 5Z-7. Strikingly, the recruitment of RIPK1 and its corresponding ubiquitination and activation in complex I was significantly reduced in $Ripk1^{K612R/K612R}$ MEFs (Fig. 2e–g). Consistent with TRADD and RIPK1 competing for binding to TNFR1 in TNFα stimulated cells as both can bind to TNFR1 through their respective DDs[37,38], the recruitment of TRADD to complex I in K612R RIPK1 MEFs was increased (Fig. 2e). The impaired recruitment of RIPK1 by K612R mutation was clearly demonstrated in cells treated with TNFα with SM-164 which promoted the degradation of cIAP1/2 and thus eliminated much of RIPK1 ubiquitination in complex I (Fig. 2f). Corresponding to the reduced levels of RIPK1 recruitment and ubiquitination, the levels of TAK1, A20, and IKKα/β recruited to complex I were reduced in RIPK1 K612R mutant MEFs (Fig. 2e). However, the recruitment of LUBAC complex, such as HOIL, HOIP, and SHARPIN, was not affected by K612R mutation.

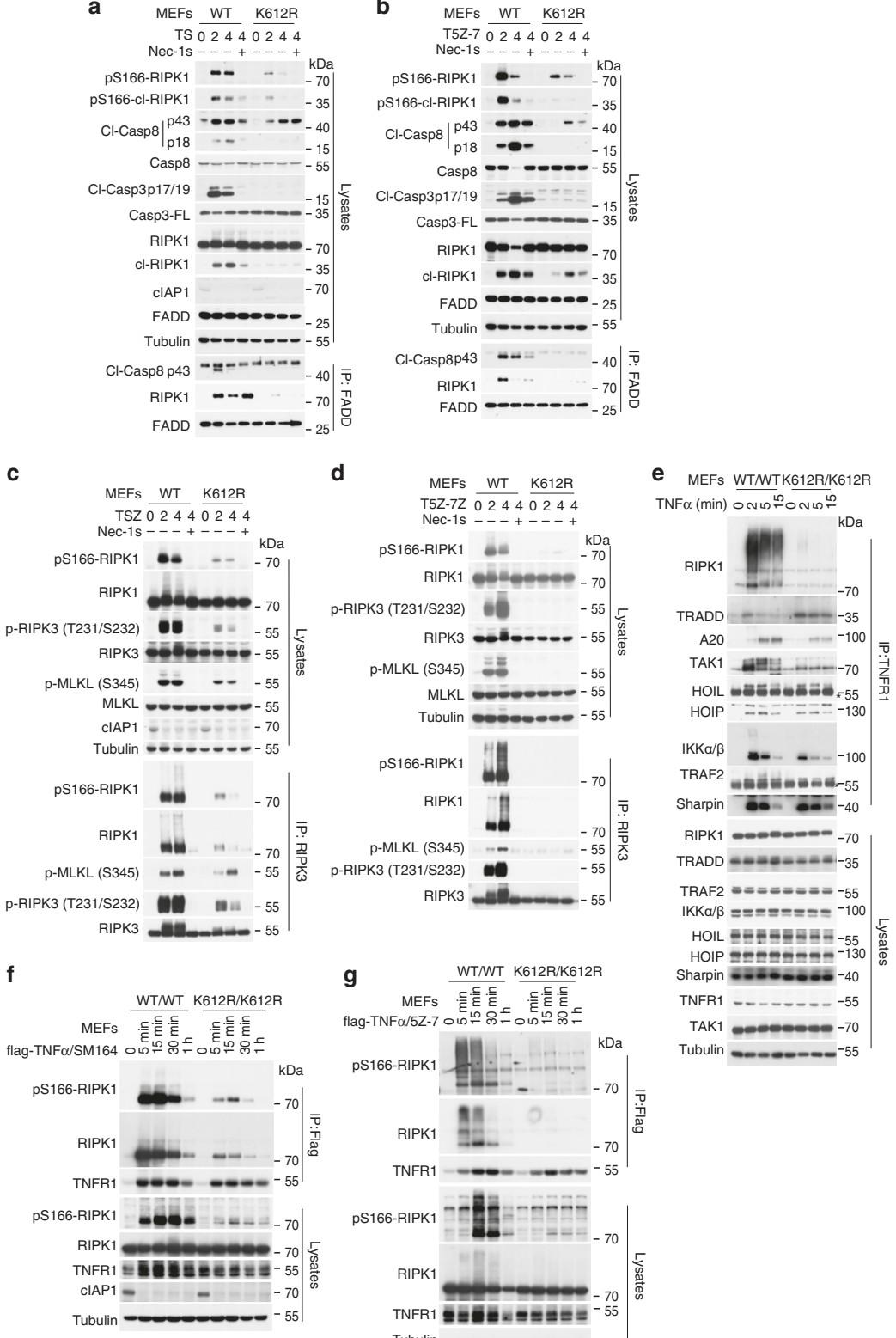

Ubiquitination and recruitment of RIPK1 into complex I is involved in mediating TNFα-induced NF-κB activation. We next investigated the role of RIPK1 K612R in mediating the activation of NF-κB and MAP kinases downstream of TNFR1. We found that the levels of p-IKKα/β, p-ERK1/2, p-JNK, and p-P38 were decreased in TNFα stimulated *Ripk1*^*K612R/K612R* MEFs and *Ripk1*^*K612R/K612R* BMDM cells compared to that of RIPK1

WT cells, corresponding to the reduced recruitment of TAK1 and IKKs into complex I as shown above (Supplementary Fig. 4a). However, p-IκBα and degradation of IκBα were comparable between RIPK1-K612R and RIPK1-WT cells. The activation of p-IKKα/β, p-IκBα, p-P38, p-ERK, and p-JNK and IκBα degradation in *Ripk1*^*K612R/K612R* MEFs was further reduced by TRADD knockout (Supplementary Fig. 4b).

**Fig. 2 K612R mutation in mRIPK1 disrupts the formation of complex I and complex II downstream of TNFR1 activation by TNFα. a, b** Complex IIa was immunoprecipitated by FADD antibodies and analyzed by western blotting in *Ripk1*[+/+] and *Ripk1*[K612R/K612R] MEFs pretreated with SM-164 (100 nM) (**a**) or 5Z-7 (200 nM) (**b**) for 2 h, and then treated with mTNFα (100 ng/ml) for different time points in the presence or absence of Nec-1s (10 μM) as indicated. Uncropped blots in the Source Data file. **c, d** Complex IIb was immunoprecipitated by RIPK3 antibodies and analyzed by western blotting in *Ripk1*[+/+] and *Ripk1*[K612R/K612R] MEFs pretreated with SM-164 (100 nM) (**c**), 5Z-7 (200 nM) (**d**) for 2 h, and then treated with mTNFα (100 ng/ml)/Z-VAD (25 μM) for indicated time. Uncropped blots in the Source Data file. **e** Complex I in *Ripk1*[+/+] and *Ripk1*[K612R/K612R] MEFs treated with mTNFα (100 ng/ml) for indicated time points was immunoprecipitated with TNFR1 antibody and analyzed by western blotting. Uncropped blots in the Source Data file. **f, g** Complex I in *Ripk1*[+/+] and *Ripk1*[K612R/K612R] MEFs pretreated with SM-164 (100 nM) (**f**) or 5Z-7 (200 nM) (**g**) for 2 h, and then treated with flag-mTNFα (100 ng/ml) for indicated time points was immunoprecipitated with Flag M2 Agarose Affinity gel and analyzed by western blotting in. Uncropped blots in the Source Data file.

We next characterized the transcription of NF-κB target genes such as TNFα, IκBα, and A20 in WT and *Ripk1*[K612R/K612R] MEFs by q-PCR. Consistently, the transcriptional levels of NF-κB target genes induced by TNFα were reduced in *Ripk1*[K612R/K612R] MEFs compared to that of WT cells, which was further reduced with TRADD knockout (Supplementary Fig. 4c).

Taken together, our results suggest that K612R mutation inhibits the recruitment and activation of RIPK1 under TNFα-mediated apoptosis and necroptosis conditions and impairs the activation of NF-κB and MAPK downstream of TNFR1. Since K612R mutation blocks both RDA and necroptosis, inhibition of RIPK1 kinase activation by K612R mutation is dominant over the impaired activation of NF-κB in RIPK1[K612R] mutant cells and leads to inhibition of cell death in TNFR1 signaling pathway.

**K612 in mRIPK1/K627 in hRIPK1 regulates the overall ubiquitination pattern of RIPK1.** To characterize the effect of K627R/K612R mutation on overall RIPK1 ubiquitination, we analyzed the Lys48 (K48), Lys63 (K63), and Met1-linked (M1) polyubiquitinations of K627R RIPK1 compared to that of WT by western blotting. We found that when expressed in 293T cells, predominantly K63 and also M1, but not K48, ubiquitination levels of hRIPK1 K627R were reduced compared to that of hRIPK1 WT (Fig. 3a–c). We next analyzed the ubiquitination linkage types of RIPK1 affected by K612R mutation in MEFs upon stimulation by TNFα or TNFα/SM-164/zVAD. We found that both M1 and K63 ubiquitinated RIPK1 species were reduced by K612R mutation in these cells stimulated by TNFα or TNFα/SM-164/zVAD, supporting the role of K612 in mRIPK1/K627 in hRIPK1 as a key ubiquitination site that can regulate the overall ubiquitination pattern of RIPK1 upon activation of TNFR1 signaling (Fig. 3d).

Next we performed a quantitative mass spectrometric analysis for the ubiquitination of WT RIPK1 and K627R RIPK1 purified from HEK293T cells. This analysis identified and quantitatively analyzed 31 ubiquitination sites on RIPK1, many of which were not previously identified. We found that the kinase domain contained surprisingly large numbers of potential ubiquitination sites, including K13, K30, K45, K49, K65, K87, K97, K105, K115, K132, K137, K140, K153, K163, K167, K185, K204, K265, K284, K302, K306, and K316; and the ubiquitination of most of these sites were reduced in K627R mutant compared to that of WT, including two of which, K265 and K302, became undetectable in K627R RIPK1 (Fig. 3e and Supplementary Table 1). A number of potential ubiquitination sites were also found in the DD or close to the DD: including K565, K571, K585, K596, K604, K634, K642, and K648. In contrast to that the ubiquitination sites found in the kinase domain, the levels of ubiquitination in some of these DD-sites, including K596, K604, K634, and K648, were significantly increased (Fig. 3e and Supplementary Table 1). These results suggest that ubiquitination of K627 localized in DD is a key ubiquitination site that can regulate the overall ubiquitination pattern of RIPK1.

We further investigated whether K612R mutation could affect the ubiquitination of endogenous RIPK1. We immunoprecipitated endogenous RIPK1 from RIPK1-WT-MEFs and RIPK1-K612R knockin MEFs that were treated with or without TNFα for 5 min using a validated rabbit monoclonal anti-RIPK1 antibody, and analyzed the ubiquitination patterns of WT and K612R RIPK1 by quantitative mass spectrometry. In WT MEFs under control condition without TNFα stimulation, many lysine residues in RIPK1 were already highly ubiquitinated, including 11 lysine residues in the kinase domain (K20, K30, K45, K65, K115, K137, K140, K153, K163, K167, and K307), 4 lysine residues in the intermediate domain (K376, K392, K395, and K429) and 4 lysine residues in the DD (K589, K612, K627, and K633) (Supplementary Table 2). Similar to the mass spec data on the ubiquitination changes of K627R RIPK1 expressed in 293T cells, the majority of these sites in the kinase domain and intermediate domain of endogenous K612R RIPK1 showed a reduction in ubiquitination levels under control condition (Fig. 3e, f and Supplementary Table 2). The ubiquitination levels of 15 lysine residues (out of the 19 ubiquitinated lysine residues in WT-RIPK1) on K612R–RIPK1 in K612R–RIPK1 knockin MEFs under control condition were reduced compared to that of WT RIPK1 in WT MEFs, including K20, K30, K45, K46, K115, K137, K140, K153, K163, K167, K307, K392, K395, K429, and K627 (Fig. 3f and Table 2). The ubiquitination levels of K65 and K376 on K612R–RIPK1 in RIPK1–K612R knockin MEFs under control conditions were comparable to that of WT-RIPK1 in WT MEFs; while three lysine residues in the DD (K589, K619, and K633) and one lysine residues in the kinase domain (K105) of K612R-RIPK1 in RIPK1-K612R knockin MEFs were increased compared to that of WT-RIPK1. With the majority of ubiquitination events in the kinase domain and the intermediate domain reduced while the ubiquitination of the DD increased by K612R mutation, the changes in overall ubiquitination patterns on endogenous RIPK1 by K612R mutation were similar to that of K627R RIPK1 expressed in 293T cells (Fig. 3e, f).

We further used quantitative mass spectrometry to compare the patterns of ubiquitination in the endogenous WT RIPK1 and K612R RIPK1 in WT and RIPK1–K612R knockin MEFs stimulated by TNFα for 5 min. In WT MEFs stimulated by TNFα for 5 min, the ubiquitination levels of 19 lysine residues, including K20, K45, K46, K65, K105, K137, K140, K153, K163, K167, K376, K392, K395, K429, K550, K589, K612, K619, and K633, were increased; the ubiquitination levels of four lysine residues, including K30, K115, K307, and K627, were decreased (Fig. 3f and Supplementary Table 2). In K612R–RIPK1 knockin MEFs stimulated by TNFα for 5 min, the ubiquitination levels of 17 lysine residues in K612R–RIPK1 were reduced compared to that of RIPK1 in WT MEFs, including K20, K30, K45, K46, K65, K105, K137, K140, K153, K163, K167, K376, K392, K395, K429, K550, and K619; while the ubiquitination levels of K589 were comparable to that of WT; and the ubiquitination levels of four lysine residues in K612R–RIPK1 in K612R–RIPK1 knockin

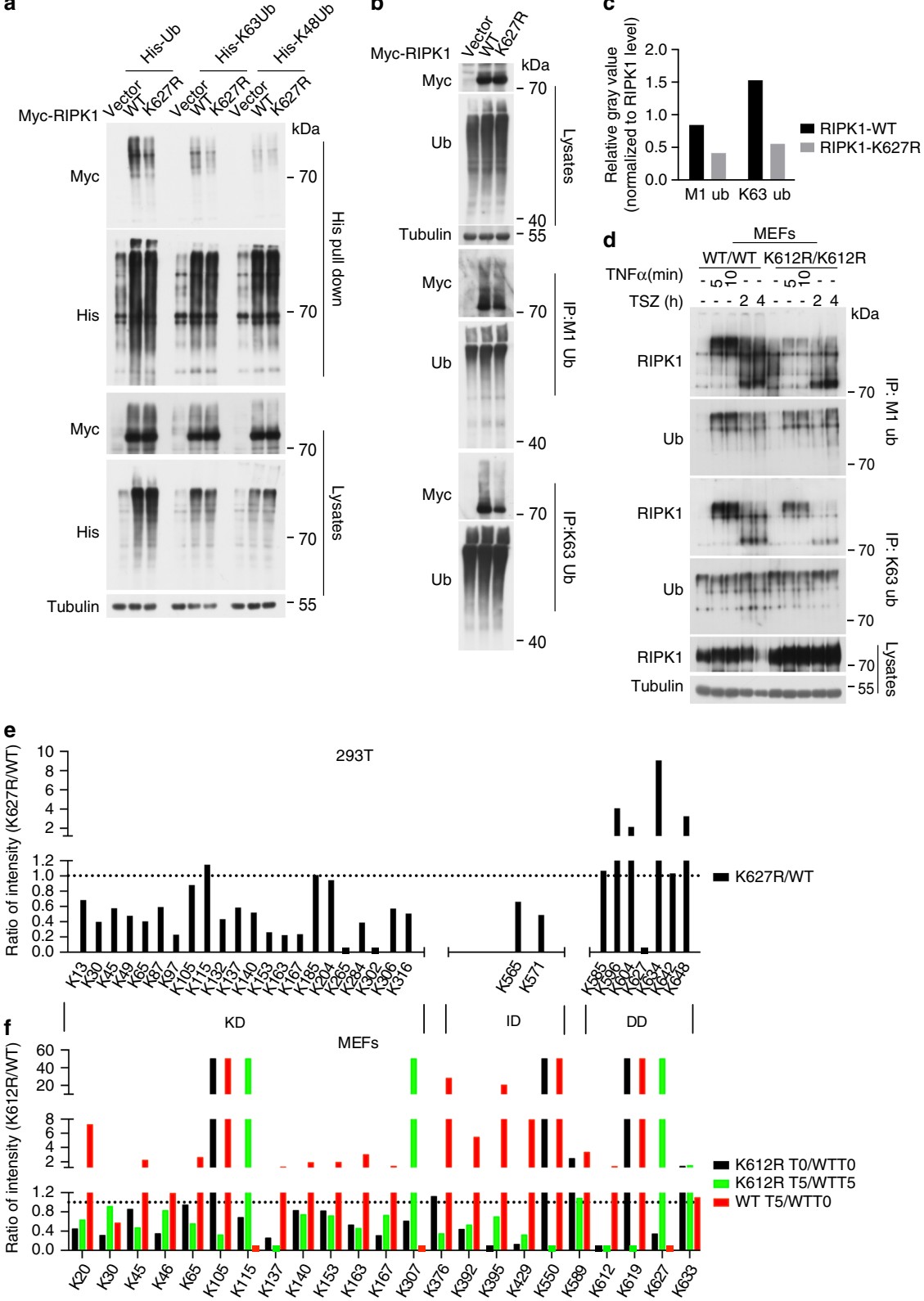

MEFs, including K115, K307, K627, and K633, were increased compared to that of WT (Fig. 3f and Supplementary Table 2). Thus, while TNFα stimulation led to a rapid increase in the ubiquitination of multiple sites over entire RIPK1, the ubiquitination levels of many of the lysine residues in K612R RIPK1 were blocked or reduced (Fig. 3f and Supplementary Table 2). Since

K612R mutation affects the levels and patterns of RIPK1 ubiquitination under both control condition and after stimulation by TNFα for 5 min, our results suggest that the ubiquitination of K612 can affect the overall pattern of RIPK1 ubiquitination in the cytoplasm under unstimulated condition as well as after TNFα stimulation which can affect its recruitment to complex I.

**Fig. 3 K612 in mRIPK1/K627 in hRIPK1 regulates the overall ubiquitination pattern of RIPK1. a** HEK293T cells were transfected with expression vectors of myc-hRIPK1-WT or myc-hRIPK1-K627R and that of His-Ub or His-K48 only Ub or His-k63 only Ub for ~18 h. The cells were lysed in 8 M urea lysis buffer and incubated with Nickle-affinity resin. The pulldown products were analyzed by western blotting using indicated antibodies. Uncropped blots in the Source Data file. **b, c** HEK293T cells were transfected with expression vectors of myc-hRIPK1-WT or myc-hRIPK1-K627R plasmids for ~18 h. The cells were lysed in 6 M urea buffer, and immunoprecipitated with linear or K63 linkage-specific anti-ubiquitin antibodies. RIPK1 ubiquitination was analyzed by western blotting with indicated antibodies (**b**). The ubiquitination levels of RIPK1-WT and RIPK1-K627R were quantified by ImageJ 1.52a and normalized to the RIPK1 levels in the corresponding cell lysates (**c**). Uncropped blots in Source Data file. **d** $Ripk1^{+/+}$ and $Ripk1^{K612R/K612R}$ MEFs were treated with mTNFα (100 ng/ml) or mTNFα (100 ng/ml)/SM-164 (100 nM)/Z-VAD (25 μM) for indicated time points. The cell lysates were immunoprecipitated in denatured condition using linear or K63 linkage-specific anti-ubiquitin antibodies and the pulldown products were analyzed by western blotting with indicated antibodies. Uncropped blots in the Source Data file. **e** Flag-RIPK1 was immunoprecipitated with Flag M2 Agarose Affinity gel in HEK293T cells transfected with expression vectors of flag-hRIPK1-WT or flag -hRIPK1-K627R for ~18 h, and then trypsin-digested and subjected to enrichment of diGly peptides. The peptides with diGly remnant were identified and quantified by mass spectrometry. The intensity of each ubiquitinated site quantified is shown in the Supplementary Table 1. The ratio of each ubiquitinated site (K) in K627R RIPK1/WT RIPK1 was plotted. The ubiquitination ratios of each site (K627R/WT) are shown in black bars. **f** RIPK1 was immunoprecipitated with a rabbit monoclonal anti-mRIPK1 antibody from $Ripk1^{+/+}$ and $Ripk1^{K612R/K612R}$ MEFs untreated (T0) or treated with mTNFα (100 ng/ml) for 5 min (T5). The ubiquitination patterns of endogenous RIPK1 were analyzed by quantitative mass spectrometry. The intensity of each ubiquitinated site quantified is shown in the Supplementary Table 2. The ratio of each ubiquitinated site (K) in T0 and T5 conditions was plotted as indicated, the value 50 indicates N.D.

**$Ripk1^{K612R/K612R}$ mice develop adult-onset intestinal inflammation and splenomegaly.** While $Ripk1^{K612R/K612R}$ mutant mice were born with normal Mendelian ratios (Fig. 4a), their growth was severely retarded (Fig. 4b). Western blotting analysis of different tissues demonstrated that RIPK1 protein levels in multiple tissues, such as spleen, thymus and intestine, were comparable between WT and $Ripk1^{K612R/K612R}$ mice at newborn or age of 3 weeks; however, they were strikingly decreased in $Ripk1^{K612R/K612R}$ mice compared with that of WT mice at the age of 6 weeks (Fig. 4c, d). We investigated if RIPK1 moved into the insoluble fraction in tissues of $Ripk1^{K612R/K612R}$ mice at 20 weeks of age and found that RIPK1 protein levels in both soluble and insoluble fraction were decreased in intestines of aged $Ripk1^{K612R/K612R}$ mice (Supplementary Fig. 5a).

At 8 weeks of age and older, $Ripk1^{K612R/K612R}$ mice developed diarrhea with altered stool consistency. The levels of pro-inflammatory cytokines, such as TNFα and interleukin-6 (IL-6), were increased in the sera of $Ripk1^{K612R/K612R}$ mice compared to that of WT (Fig. 4e). Histo-pathological analysis showed characteristics of inflammatory bowel disease with moderately reduced colon length, serious colonic wall thickening, increased infiltration of macrophages (F4/80+), neutrophil (GR-1) and hyperproliferation of colon epithelial cells (IECs) (Ki67+) in colon, but ileum was less affected (Fig. 4f, g and Supplementary Fig. 5b). In contrast to that of $Ripk1^{IEC-KO}$ or $Fadd^{IEC-KO}$ mice[39,40], no extensive loss of Paneth cells or goblet cells was observed in $Ripk1^{K612R/K612R}$ mice at 8 weeks of age, as indicated by PAS/AB or lysozyme staining (Fig. 4g and Supplementary Fig. 5b). Dramatic increased inflammatory cytokines, including increased mRNA levels of TNFα, IL-6, IL-1 β, Cxcl1, Cxcl2, and CCL2 was a key pathological feature in the colon of $Ripk1^{K612R/K612R}$ mice (Fig. 4h). Slight increases of Cxcl2 and CCL2 were found in ileum of $Ripk1^{K612R/K612R}$ mice compared to that of WT mice (Supplementary Fig. 5c). In addition, we observed a dramatically reduced level of cFLIP mRNA in colons of $Ripk1^{K612R/K612R}$ mice compared with that of WT mice (Fig. 4i). Thus, these data suggest that RIPK1 K612R mutation leads to dysregulation of RIPK1 ubiquitination, which promotes a strong inflammatory response in the large intestine of adult $Ripk1^{K612R/K612R}$ mice without extensive epithelial cell death.

$Ripk1^{K612R/K612R}$ mice also displayed progressive spleen enlargement (Fig. 4j), while the sizes of thymus and lymph nodes were normal as that of WT mice until 20 weeks of age (oldest mice observed) (Supplementary Fig. 5d). Unlike $Fadd^{-/-}Mlkl^{-/-}$ and $Fadd^{-/-}Ripk3^{-/-}$ mice which accumulate an abnormally large population of B220+CD3+ T lymphocytes in spleens, thymus, and lymph nodes of aged mice[41,42], the percentages of different immune subsets including, CD4+, CD8+, and B220+CD3+ T lymphocytes were similar in spleens, thymus, and lymph nodes of aged $Ripk1^{K612R/K612R}$ mice (Supplementary Fig. 5e, f). Splenomegaly and large increases of myeloid-derived suppressor cells (MDSCs) in spleens and bone marrows were pathological features of various mouse colitis models including combined IEC and myeloid A20 deficiency induced intestinal inflammation, mouse colitis models induced by 2,4,6-trinitrobenzene sulfonic acid (TNBS) or dextran sulfate sodium (DSS)[43,44]. MDSCs in mice are defined as a heterogeneous population of cells that simultaneously express Gr1 and CD11b. MDSCs can be used as a biomarker for disease activity of IBD. Consistently, the percentages of CD11b+ and Gr1+CD11b+ cells were significantly increased in the spleens and bone marrow of $Ripk1^{K612R/K612R}$ mice compared to that of WT (Fig. 4k and Supplementary Fig. 5e, f). This observation suggests that there is an extensive expansion of myeloid-derived suppressor cell (MDSC) population in $Ripk1^{K612R/K612R}$ mice.

**Antibiotics rescues spontaneous gut inflammation of RIPK1 K612R mutant mice.** Alterations of commensal bacteria contribute to experimental gut inflammation and in human IBD[45,46]. To investigate the contribution of microbiota to colitis in K612R mice, we treated littermates of WT and $Ripk1^{K612R/K612R}$ mice with a cocktail of broad-spectrum antibiotics for 4 weeks starting shortly after weaning from 4 weeks of age. We found that the body weight, increased levels of TNFα and IL-6 in sera, spleen weights, and colon length of $Ripk1^{K612R/K612R}$ mice were normalized by antibiotic treatment (Fig. 5a–d). Interestingly, the reduced levels of RIPK1, e.g., in colon and spleens of K612R mice, were restored by antibiotics (Fig. 5e). Accordingly, the infiltration of macrophages (F4/80+), neutrophil (GR-1+) and hyperproliferation of colon epithelial cells (IECs) (Ki67+) were also inhibited by antibiotics treatment (Fig. 5f). The production of proinflammatory cytokines was dramatically reduced and cFLIP mRNA levels of colon were normalized in antibiotic treated K612R mice (Fig. 5g). Moreover, the accumulation of Gr1+CD11b+ cells in spleen and bone marrow of $Ripk1^{K612R/K612R}$ mice were prevented by antibiotic treatment (Fig. 5h). These findings suggest that the spontaneous gut inflammation and splenomegaly in $Ripk1^{K612R/K612R}$ mice are predominantly induced by intestinal bacteria. Taken together, our results suggest that ubiquitination

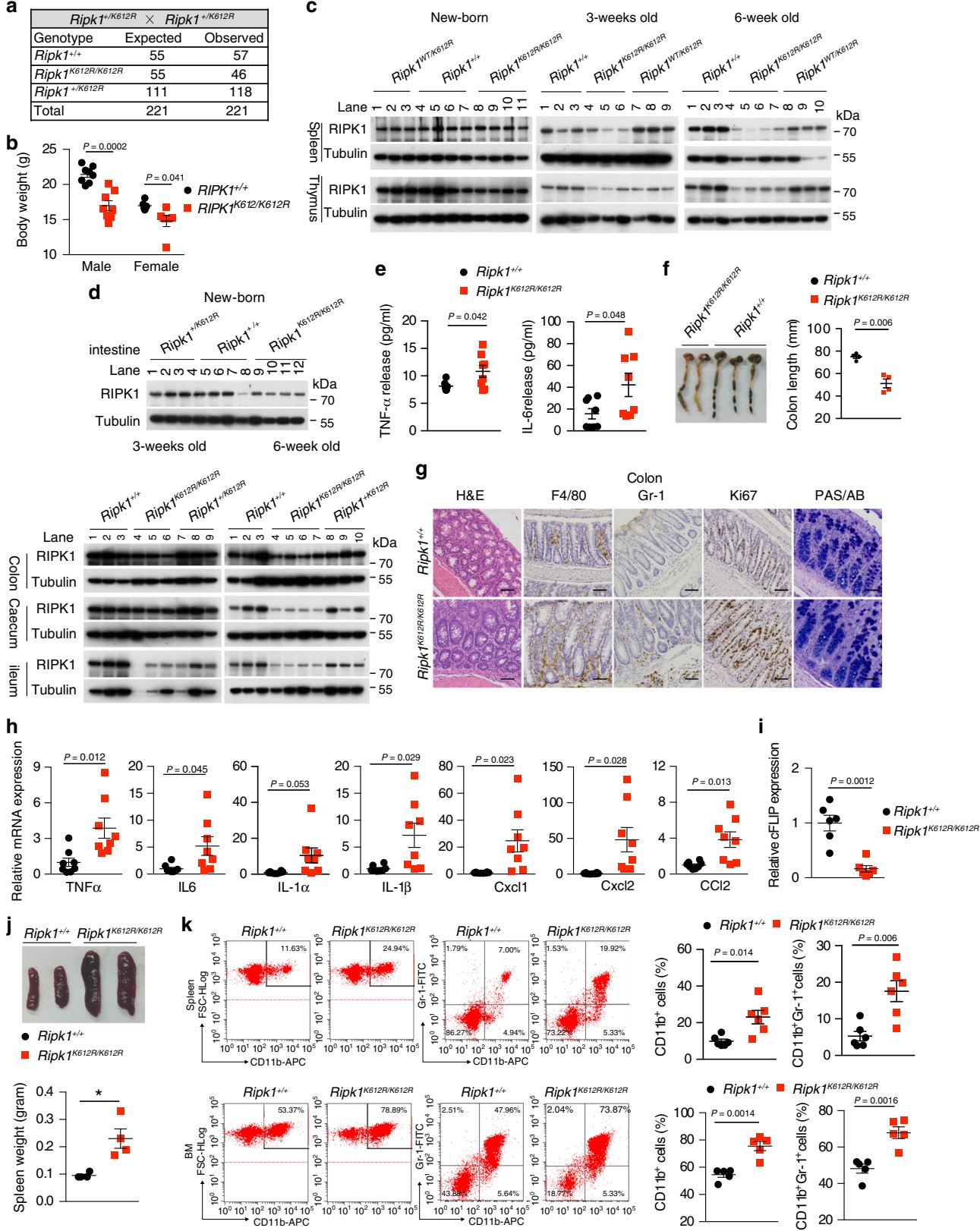

of K612/K627 RIPK1 is involved in regulating inflammatory response that controls the levels of RIPK1 in vivo.

**RIPK1 K612R mutation promotes necroptosis and caspase-1 activation mediated by TLR3/4.** The ability of antibiotics to rescue gut inflammation in *Ripk1*[K612R/K612R] mice suggests the

involvement of TLRs signaling. Activation of TLR3 and TLR4 can promote necroptosis through a RIP homotypic interaction motif-dependent association of TRIF with RIPK3 and RIPK1[47–49]. We next investigated whether RIPK1[K612R] mutation affects TLR3/ TLR4 signaling, we stimulated the BMDMs cells isolated from WT and RIPK1[K612R] mutant mice with Poly (I:C) or LPS in the

**Fig. 4 $Ripk1^{K612R/K612R}$ mice develop adult-onset intestinal inflammation and splenomegaly. a** Expected and observed frequency of genotypes in the offspring as adult from intercrosses of $Ripk1^{+/K612R}$ mice. **b** Body weight of 8-weeks-old male $Ripk1^{+/+}$ ($n = 8$) and $Ripk1^{K612R/K612R}$ ($n = 8$) mice, and female $Ripk1^{+/+}$ ($n = 6$) and $Ripk1^{K612R/K612R}$ ($n = 6$) mice. Data are presented as mean ± SEM of $n = 8$ or 6 biologically independent mice. Unpaired two-tailed Student's $t$- test. **c, d** Western blotting analysis of RIPK1 protein levels in the spleen and thymus (**c**) and intestine (**d**) from indicated mice at different ages. Uncropped blots in Source Data. **e** IL-6 and TNFα levels in the sera of $Ripk1^{+/+}$ ($n = 8$) and $Ripk^{K612R/K612}$ ($n = 8$) (8–9 weeks-old) mice determined by ELISA. Data are presented as mean ± SEM of $n = 8$ biologically independent mice. Unpaired two-tailed Student's $t$-test. **f** Representative images of large intestine and colon length of $Ripk1^{+/+}$ ($n = 4$) and $Ripk1^{K612R/K612R}$ mice ($n = 4$, 8 weeks old). Data are presented as Mean±SEM of $n = 4$ biologically independent mice. Unpaired two-tailed Student's $t$-test. **g** Haematoxylin & Eosin staining for general histology, PAS/AB staining for goblet cells and immunohistochemical staining of F4/80, GR-1 and Ki67 in the sections of colon from $Ripk1^{+/+}$ ($n = 4$) and $Ripk1^{K612R/K612R}$ ($n = 4$) mice (8–9 weeks old) (Scale bars, 50 μm). **h** QRT-PCR analysis of cytokine and chemokine expression in the colon of $Ripk1^{+/+}$ ($n = 8$) and $Ripk1^{K612R/K612R}$ ($n = 8$) (8–9 weeks old). Data are presented as mean ± SEM of $n = 8$ biologically independent mice. Unpaired two-tailed Student's $t$-test. **i** QRT-PCR analysis of cFLIP expression in the colon of $Ripk1^{+/+}$ ($n = 6$) and $Ripk1^{K612R/K612R}$ ($n = 6$) (8–9 weeks old). Data are presented as mean ± SEM of $n = 6$ biologically independent mice. Unpaired two-tailed Student's $t$-test. **j** Representative images of spleens and spleen weight of $Ripk1^{+/+}$ mice ($n = 3$) and $Ripk1^{K612R/K612R}$ mice ($n = 3$) (8–9 weeks old). **k** The populations of $CD45^+CD11b^+GR-1^+$ cells in the spleens of $Ripk1^{+/+}$ ($n = 6$) and $Ripk1^{K612R/K612R}$ ($n = 6$) mice, and bone marrow of $Ripk1^{+/+}$ ($n = 5$) and $Ripk1^{K612R/K612R}$ ($n = 5$) mice (8 weeks old). Dot plot analysis was performed to visualize the percentages of indicated cell populations. Data are presented as mean ± SEM biologically independent mice. Unpaired two-tailed Student's $t$-test.

presence or absence of zVAD.fmk. Surprisingly, in contrast to the resistance offered by K612R mutation to TNFR1-mediated apoptosis and necroptosis, RIPK1$^{K612R}$ BMDMs showed increased sensitivity to cell death induced by LPS and Poly (I:C) alone as well as in the presence of Z-VAD.fmk compared to that of WT, and the cell death was effectively inhibited by Nec-1s, RIPK3 inhibitor GSK'872 (Fig. 6a). RIPK1$^{K612R}$ mutant MEFs also showed increased sensitivity to necroptosis induced by Poly (I:C) plus Z-VAD, which was inhibited by Nec-1s and GSK'872 (Supplementary Fig. 6a).

We next examined whether RIPK1$^{K612R}$ affected necrosome formation induced by LPS only or LPS/Z-VAD. LPS stimulation alone, a condition not sufficient to induce the activation of necroptosis in WT BMDMs, induced robust activation of necroptosis of RIPK1$^{K612R}$ BMDMs as marked by p-S166 RIPK1, p-T231/S232 RIPK3, and p-S345 MLKL and the formation of complex IIb including activated RIPK1, RIPK3 and MLKL (Fig. 6b). Thus, K612R mutation sensitizes BMDMs to necroptosis in response to LPS stimulation. TLR3/TLR4 signaling, which was not sufficient to induce necroptosis of WT BMDMs, induced RIPK1 kinase-dependent and RIPK3-dependent necroptosis in RIPK1$^{K612R}$ mutant BMDMs.

Furthermore, inhibition of caspase activation upon LPS/Z-VAD stimulation in RIPK1$^{K612R}$ BMDMs resulted in dramatic increases of p-S166 RIPK1, p-RIPK3, p-MLKL, and RIPK3/RIPK1/MLKL complex formation compared with that in WT BMDMs (Fig. 6c). These results suggest that RIPK1$^{K612R}$ mutation promotes RIPK1 activation and necroptosis in BMDMs mediated by TLR3/4 signaling.

We next investigated the role of RIPK1 K612 in caspase-1 activation and IL-1β secretion triggered by TLR3 and TLR4 activation. We found that RIPK1$^{K612R}$ had no effects on LPS or Poly (I:C)-induced NF-κB and MAP kinases in BMDM cells (Supplementary Fig. 6b). Extracellular ATP, acting at plasma membrane purinergic P2 receptor P2X7 receptor (P2X7R) subtype, is required to drive NLRP3 inflammasome activation, caspase-1 cleavage, and processing and release of IL-1β in WT cells[50,51]. Interestingly, the secretion of IL-1β, increased cleavage and secretion of both caspase-1 and IL-1β compared with that of WT cells were observed in RIPK1$^{K612R}$ BMDMs stimulated with LPS in the absence of extracellular ATP; while TNFα release was not affected by RIPK1$^{K612R}$ mutation (Fig. 6d, e). Since LPS stimulated synthesis of pro-IL-1β was not affected by K612R (Fig. 6e), this result strongly suggests that RIPK1$^{K612R}$ BMDMs are also sensitized to caspase-1-mediated release of mature IL-1β. Similar to that of LPS, Poly (I:C) stimulation also

dose-dependently promoted IL-1β release in RIPK1$^{K612R}$ BMDMs (Supplementary Fig. 6c).

The increased IL-1β release in RIPK1$^{K612R}$ mutant BMDMs stimulated by LPS was inhibited by RIPK1 kinase inhibitor (Nec-1s) and RIPK3 inhibitor (GSK'872). Treatment of GSK'872 inhibited phosphorylation of MLKL, the downstream target of RIPK3[26], while p-S166 RIPK1 was increased (Supplementary Fig. 6d, e). Thus, RIPK1$^{K612R}$ mutation sensitizes TLRs ligand-stimulated BMDMs to caspase-1 activation mediated by RIPK1 kinase and RIPK3 kinase activities which in turn promotes IL-1β release in response to LPS and Poly (I:C) stimulation alone, which does not happen in WT cells.

We next tested the role of RIPK3 in the inflammatory phenotype upon genetic knockout of $Ripk3$. The reduced body weight of $Ripk1^{K612R/K612R}$ mice was rescued in $Ripk1^{K612R/K612R};$ $Ripk3^{-/-}$ mice (Supplementary Fig. 7a). The spleen weight of $Ripk^{K612R/K612R}; Ripk3^{-/-}$ mice was also normalized compared with $Ripk^{K612R/K612R}$ mice and $Ripk1^{WT/WT}; Ripk3^{+/+}$ mice (Supplementary Fig. 7b). The accumulation of Gr1+ CD11b+ populations in spleens and bone marrows in $Ripk^{K612R/K612R}$ mice was effectively rescued by Ripk3 deficiency (Supplementary Fig. 7c). In BMDMs, genetic RIPK3 deficiency effectively inhibited cell death induced by LPS and Poly (I:C) alone as well as in the presence of Z-VAD.fmk (Fig. 6f), and abolished RIPK1 S166 phosphorylation and necroptosis induced by LPS/Z-VAD in both RIPK1 WT and K612R mutant BMDMs, indicating that the scaffold function of RIPK3 may mediate RIPK1 activation in K612R mutant BMDMs upon TLRs signaling, which in turn promote necroptosis (Fig. 6g). Furthermore, the increased caspase-1 activation and IL-1β release in RIPK1$^{K612R}$ mutant BMDMs stimulated by LPS was completely inhibited by RIPK3 deficiency (Fig. 6h, i). Taken together, these results indicate that RIPK1$^{K612R}$ mutation leads to disinhibition of RIPK3, which in turn promotes caspase-1 activation and IL-1β cleavage upon LPS stimulation in BMDMs.

The large intestines of $Ripk1^{K612R/K612R}; Ripk3^{-/-}$ mice were also morphologically normalized, suggesting that gut inflammation in $Ripk1^{K612R/K612R}$ mice was substantially ameliorated by Ripk3 deficiency (Supplementary Fig. 7d); while RIPK3 deficiency had no effect on RIPK1 protein expression levels in the colons of $Ripk1^{K612R/K612R}$ mice (Supplementary Fig. 7e). The circulating level of interleukin-6 (IL-6) was significantly reduced by Ripk3 deficiency in $Ripk1^{K612R/K612R}$ mice (Supplementary Fig. 7f), supporting a pro-inflammatory role of RIPK3 in $Ripk^{K612R/K612R}$ mice. Q-PCR analysis of inflammatory gene expression in colon showed moderate downregulation of IL-1β,

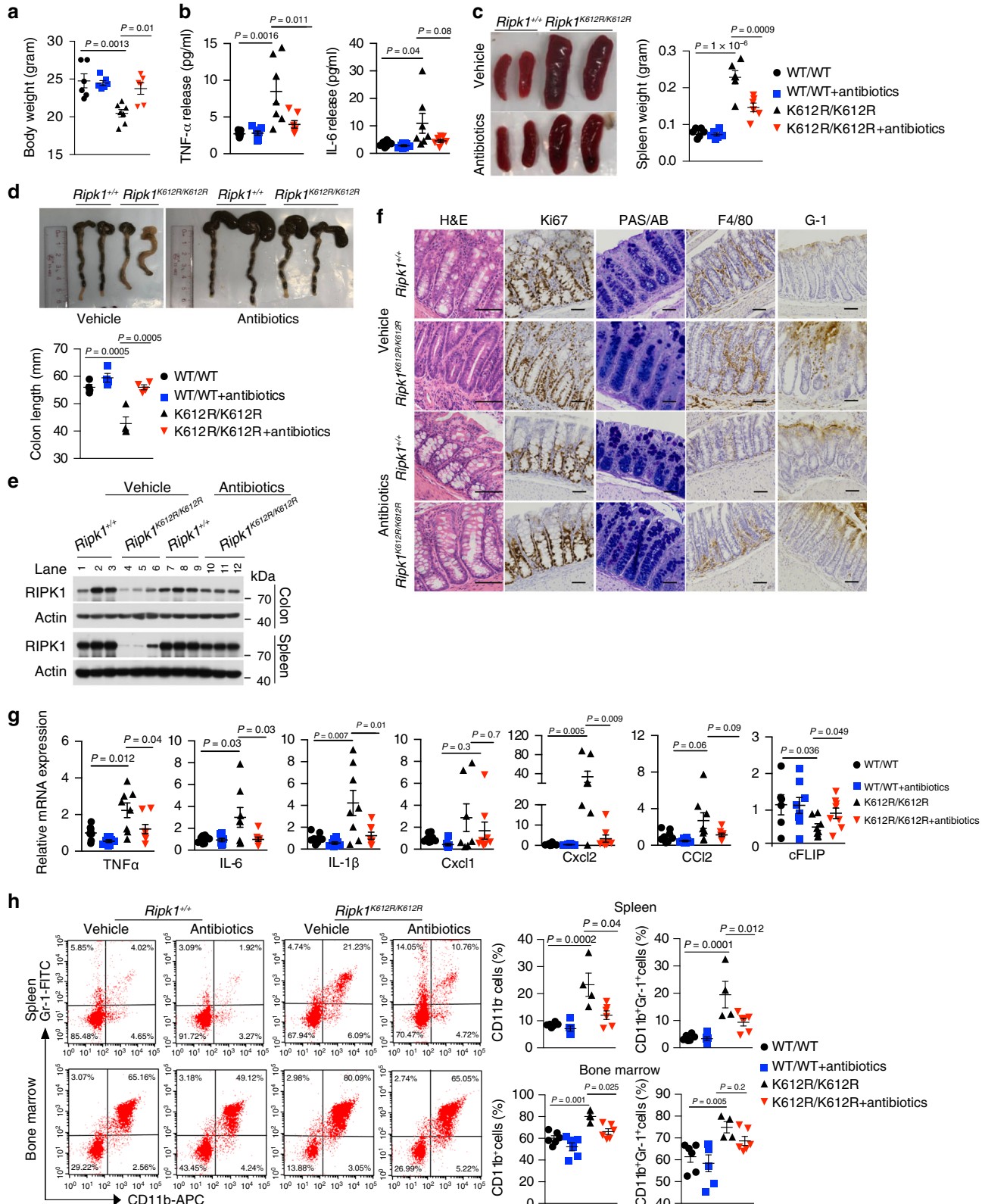

TNF-α, IL-6, Cxcl1, Cxcl2, and Ccl2 in $Ripk^{K612R/K612R}$; $Ripk3^{−/−}$ mice compared with $Ripk^{K612R/K612R}$ mice but did not reach statistical significance (Supplementary Fig. 7g). The reduced mRNA levels of cFLIP in colon section of $Ripk1^{K612R/K612R}$ mice were also not changed by Ripk3 deletion, suggesting the persistent defect of NF-κB pathway in $Ripk1^{K612R/K612R}$; $Ripk3^{−/−}$ mice

(Supplementary Fig. 7h). Taken together, these data suggest that RIPK3 deficiency protects against the hyperactivation of myeloid cells which contributes to intestinal inflammation of $Ripk1^{K612R/K612R}$ mice; however, the resistant defect in NF-κB pathway may explain the remaining intestinal inflammation in $Ripk^{K612R/K612R}$; $Ripk3^{−/−}$ mice.

**Fig. 5 Antibiotic treatment rescues the spontaneous gut inflammation in _Ripk1$^{K612R/K612R}$_ mutant mice. a** Body weight of 8 weeks-old male _Ripk1$^{+/+}$_ mice treated with antibiotics ($n = 6$) or vehicle ($n = 6$), and male _Ripk1$^{K612R/K612R}$_ mice treated with antibiotics ($n = 6$) or vehicle ($n = 6$) for 4 weeks. Data are presented as mean ± SEM of $n = 6$ biologically independent mice. One-way ANOVA with Dunnett's multiple comparison test. **b** IL-6 and TNFα level in sera of 8 weeks-old _Ripk1$^{+/+}$_ mice treated with antibiotics ($n = 7$) or vehicle ($n = 7$), and _Ripk1$^{K612R/K612R}$_ mice treated with antibiotics ($n = 7$) or vehicle ($n = 7$) for 4 weeks. Data are presented as Mean±SEM of $n = 7$ biologically independent mice. One-way ANOVA with Dunnett's multiple comparison test. **c** Representative images of spleens and spleen weight of 8–9 weeks-old _Ripk1$^{+/+}$_ mice treated with antibiotics ($n = 6$) or vehicle ($n = 6$), and _Ripk1$^{K612R/K612R}$_ mice treated with antibiotics ($n = 6$) or vehicle ($n = 6$) for 4 weeks. Data are presented as Mean±SEM of $n = 8$ or 7 biologically independent mice. One-way ANOVA with Dunnett's multiple comparison test. **d** Representative images of the large intestine and colon length of 8 weeks-old male _Ripk1$^{+/+}$_ mice treated with antibiotics ($n = 4$) or vehicle ($n = 4$), and male _Ripk1$^{K612R/K612R}$_ mice treated with antibiotics ($n = 4$) or vehicle ($n = 4$) for 4 weeks. Data are presented as mean ± SEM of $n = 4$ biologically independent mice. One-way ANOVA with Dunnett's multiple comparison test. **e** Western blotting analysis of RIPK1 protein levels of the colon and spleen from indicated mice treated with antibiotics or vehicle for 4 weeks as in (d). Uncropped blots in the Source Data file. **f** Haematoxylin & Eosin staining, PAS/AB staining, and immunohistochemical staining of F4/80, GR-1, and Ki67 in the sections of colon from 8-weeks-old _Ripk1$^{+/+}$_ mice treated with antibiotics ($n = 3$) or vehicle ($n = 3$), and _Ripk1$^{K612R/K612R}$_ mice treated with antibiotics ($n = 3$) or vehicle ($n = 3$) for 4 weeks (Scale bars, 50 μm). **g** QRT-PCR analysis of cytokine and cFLIP expression in the colon of 8-weeks-old _Ripk1$^{+/+}$_mice treated with antibiotics ($n = 8$ or 6) or vehicle ($n = 8$), and _Ripk1$^{K612R/K612R}$_ mice treated with antibiotics ($n = 8$) or vehicle ($n = 8$) for 4 weeks. Data are presented as mean ± SEM of $n = 8$ or 6 biologically independent mice. One-way ANOVA with Dunnett's multiple comparison test. **h** The populations of CD45$^+$CD11b$^+$GR-1$^+$ cells in the spleens and bone marrow of 8-weeks-old _Ripk1$^{+/+}$_ mice treated with antibiotics ($n = 6$) or vehicle ($n = 6$), and _Ripk1$^{K612R/K612R}$_ mice treated with antibiotics ($n = 6$) or vehicle ($n = 4$) for 4 weeks. Dot plot analysis was performed to visualize the percentages of indicated cell populations. Data are presented as mean ± SEM of $n = 6$ or 4 biologically independent mice. One-way ANOVA with Dunnett's multiple comparison test.

**RIPK1 K612R/K627R disrupts DD-mediated homodimerization and heterodimerization**. Since K612 and K627 are localized in the DDs of murine and human RIPK1, respectively, which can mediate homodimerization with itself, as well as heterodimerization with other DD domain containing proteins, including TNFR1, TRADD, or FADD[52]. We investigated the effect of K627R/K612R mutations on RIPK1 homodimerization and heterodimerization. We found that RIPK1 K627R mutation dramatically inhibited the interactions between full length RIPK1 and truncated RIPK1 DD only (Fig. 7a).

We examined the effect of RIPK1 K627R in the context of full length RIPK1. We found that the interaction of K627R RIPK1 with WT RIPK1 was reduced to approximately that of RIPK1-ΔDD (Fig. 7b). To eliminate the contribution of RHIM in the intermediate domain, which can promote RIPK1 aggregation when overexpressed, we mutated the IQIG in the RHIM to 4A. We found that RIPK1-4A-ΔDD mutant was unable to bind to WT RIPK1 while the binding of RIPK1-4A-K627R with WT RIPK1 was further reduced compared to that of RIPK1-4A (Fig. 7b). These results suggest that K627 in hRIPK1 is important for RIPK1 homodimerization.

We next characterized the role of K612 in mediating mRIPK1 dimerization by establishing an inducible RIPK1 dimerization system expressing different RIPK1 variants fused with two copy of FKBP$^{F36V}$ at the N termini. Induction of dimerization by the addition of AP20187 and zVAD.fmk efficiently induced RIPK1 activation and necroptosis in a manner dependent on RIPK1 and RIPK3 kinase activity, which was inhibited by both Nec-1s and GSK'872. Interestingly, the resistance of RIPK$^{K612R}$ mutant cells could be overcome by forced RIPK1 dimerization (Fig. 7c). Induced dimerization of RIPK1$^{K612R}$ robustly activated RIPK1 (p-S166) which was inhibited by Nec-1s and also p-MLKL, which was inhibited by GSK'872 and Nec-1s (Fig. 7d). These results suggest that RIPK1$^{K612R}$ mutation inhibits TNF-induced cell death by blocking RIPK1 homodimerization mediated by the DD, which could be overcome by forced dimerization.

We next characterized the effect of K627R mutation on the heterodimerization of RIPK1 with TNFR1, TRADD, and FADD. We found that the interactions of RIPK1 with the DDs of TNFR1, TRADD, and FADD were strongly inhibited by K627R mutation (Fig. 7e). Thus, K612/K627 is critical for mediation of RIPK1 interactions with itself as well as other DD containing proteins. The role of K612/K627 in both homodimerization and heterodimerization of RIPK1 provides an opportunity for us to assess the function of RIPK1 DD in different signaling pathways.

To explore the mechanism by which RIPK1$^{K612R}$ mutation promotes RIPK3 activation in TLRs signaling, we expressed dimerizable RIPK3 fused with 2×FKBP$^{F36V}$ at the N terminus in RIPK1$^{WT}$ and RIPK1$^{K612R}$ mutant MEFs. Induction of RIPK3 dimerization by the addition of AP20187 and zVAD.fmk was more effective in inducing necroptosis in K612R mutant MEFs than that of WT MEFs and the cell death was inhibited by RIPK3 kinase inhibitor GSK'872 but not Nec-1s (Fig. 7f, g). Furthermore, the activation of RIPK3 dimerization in K612R MEFs enhanced the p-RIPK3 and p-MLKL compared to that of WT MEFs and led to the activation of p-S166-RIPK1, which was minimum in WT MEFs (Fig. 7g). These data suggest that RIPK1 K612R mutation may lead to a disinhibition on RIPK3.

Since K612R mutation affected the binding of RIPK1 with other DDs, we hypothesized that the increased sensitivity of RIPK1$^{K612R}$ MEFs and RIPK1$^{K612R}$ BMDMs to LPS and Poly (I: C) might be due to defects in interacting with other DD-containing proteins such as FADD and TRADD. To test this possibility, we deleted FADD and TRADD individually by means of CRISPR/Cas9 technology in MEF cells, to investigate their effects on the sensitization of RIPK1$^{K612R}$ mutation to RIPK3 activation in response to TLRs signaling. The activation of RIPK1 and RIPK3 and necroptosis was induced by forced dimerization of RIPK3 in WT MEFs, which was further enhanced by FADD knockout. However, the necroptosis of RIPK1$^{K612R}$ MEFs and the activation of p-RIPK1, p-RIPK3, and p-MLKL induced by RIPK3-dimerization were not further affected by FADD knock-out (Fig. 7h, i). Moreover, FADD deficiency also increased Poly (I:C)/Z-VAD induced p-RIPK1, p-RIPK3, p-MLKL, and necroptosis in RIPK1$^{WT}$ MEF cells, but showed limited effects on that of RIPK1$^{K612R}$ MEF cells (Supplementary Fig. 8a, b). In contrast, TRADD deficiency did not affect RIPK3 dimerization-induced necroptosis or TLR3-mediated necroptosis in RIPK1$^{K612R}$ mutant cells (Supplementary Fig. 8c–f). These data suggest that the interaction of RIPK1 with FADD mediated by their DDs involving RIPK1 K612 may be important in inhibiting RIPK1 and RIPK3 activation and necroptosis activated by TLRs signaling. This is in contrast to that of the interaction of RIPK1 with TNFR1 mediated by their respective DDs, which is important in promoting the activation of RIPK1 and necroptosis activated by TNFα.

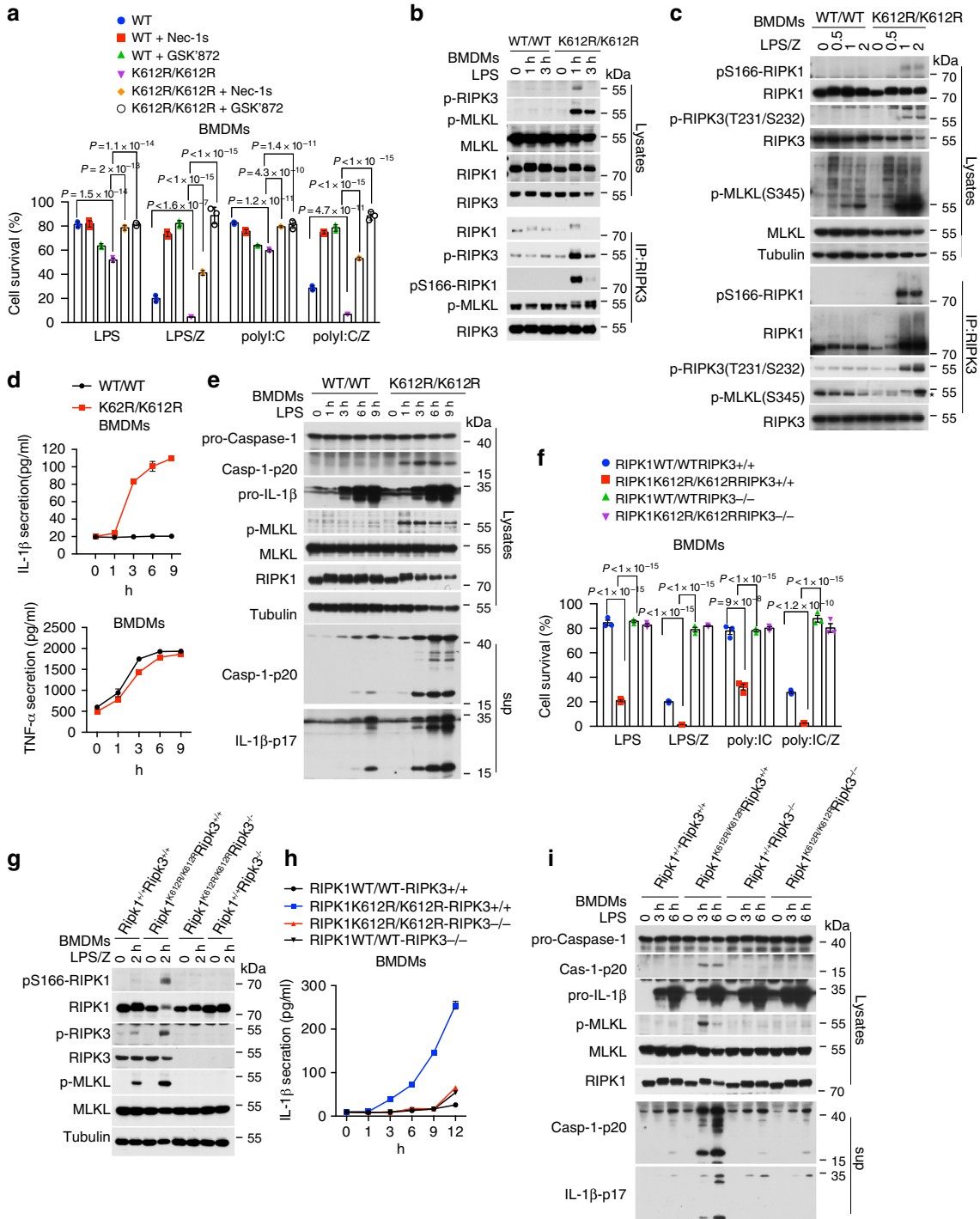

**Fig. 6 RIPK1 K612R mutation promotes necroptosis and caspase-1 activation mediated by TLR3/4. a** Primary *Ripk1*[+/+] and *Ripk1*[K612R/K612R] BMDMs were pretreated with Nec-1s (10 μM), GSK'872 (10 μM) or vehicle in the presence or absence of Z-VAD (25 μM) for 30 min respectively, and then treated with Poly (I:C) (20 μg/ml) or LPS (50 ng/ml) as indicated for 10 h. Cell survival was measured by CellTiterGlo. Data are presented as mean ± SEM of $n = 3$ biologically independent samples. Two-way ANOVA with Bonferroni's multiple comparison test. **b**, **c** Complex IIb was isolated by RIPK3 antibody and analyzed by western blotting in primary BMDMs isolated from indicated mouse strains pretreated with Z-VAD (25 μM) or vehicle for 30 min, and then treated with LPS (50 ng/ml) for indicated time points. Uncropped blots in the Source Data file. **d** Quantification of IL-1β and TNFα in the cultural supernatant from primary *Ripk1*[+/+] and *Ripk1*[K612R/K612R] BMDMs treated with LPS (50 ng/ml) for indicated time points by ELISA. **e** Western blotting analysis of the cell lysates and cultural supernatant of *Ripk1*[+/+] and *Ripk1*[K612R/K612R] primary BMDMs treated with LPS (100 ng/ml) for indicated time points. Uncropped blots in the Source Data file. **f** Primary BMDMs isolated from indicated mouse strains were treated with LPS (50 ng/ml) plus Z-VAD (25 μM) or Poly (I:C) (20 μg/ml) or plus Z-VAD (25 μM) as indicated for 10 h. Cell survival was measured by Cell TiterGlo. Data are presented as Mean±SEM of $n = 3$ biologically independent samples. Two-way ANOVA with Bonferroni's multiple comparison test. **g** Western blotting analysis of primary BMDMs isolated from indicated mouse strains were treated with LPS (50 ng/ml) plus Z-VAD (25 μM) for indicated time points. Uncropped blots in the Source Data file. **h**, **i** Quantification of IL-1β in cultural supernatant by ELISA (**h**) and western blotting analysis of cell lysates and cultural supernatant (**i**) of primary BMDMs with indicated genotypes treated with LPS(50 ng/ml) for indicated time points. Uncropped blots in the Source Data file.

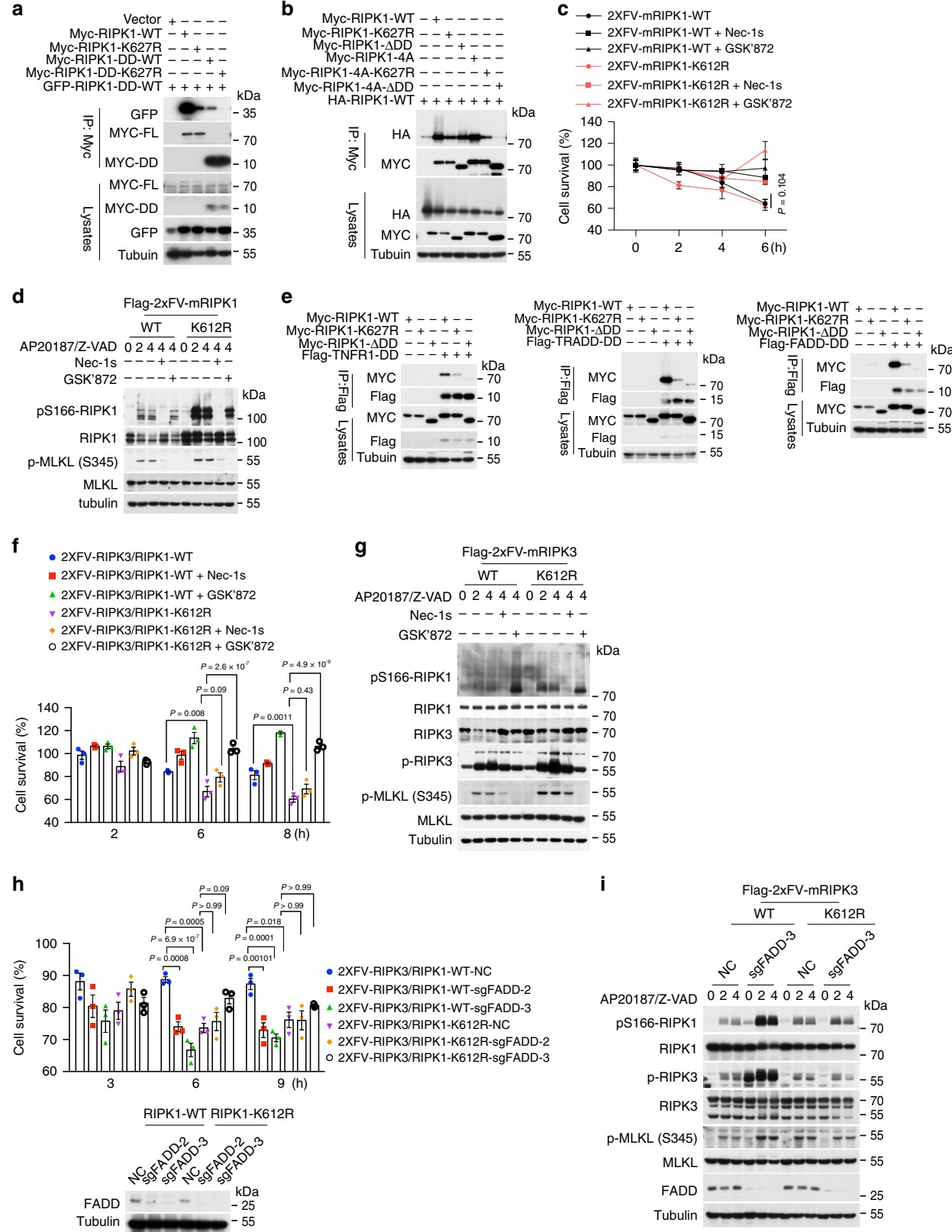

## Discussion

In the present study, we show that K627 and K612 in human and murine RIPK1 control the overall ubiquitination pattern of RIPK1, and are critical in mediating both homodimerization as well as the heterodimerization of RIPK1 with TNFR1, TRADD, and FADD. Using K612R knockin mice, we dissected the

function of RIPK1-DD and RIPK1 ubiquitination in TNFR1 and TLRs signaling. We show that reduced interaction of RIPK1-DD with other DD-containing proteins leads to opposite consequences in cells activated by TNFR1 and TLRs signaling. In TNFR1 signaling pathway, inhibiting the recruitment of RIPK1 to complex I associated with TNFR1 by K612R mutation blocks the

**Fig. 7 RIPK1 K612R/K627R mutation disrupts DD-mediated homodimerization and heterodimerization. a, b** Immunoprecipitation and western blotting analysis of RIPK1 interaction in HEK293T cells cotransfected with expression vectors of myc-hRIPK1-WT-FL, myc-RIPK1-hK627R-FL, myc-hRIPK1-DD-WT, myc-hRIPK1-DD-K627R, and GFP-hRIPK1-DD-WT (**a**) or that of myc-hRIPK1-WT, myc-hRIPK1-K627R, myc-hRIPK1-ΔDD, myc-hRIPK1-WT-4A, myc-hRIPK1-K627R-4A, myc-hRIPK1-ΔDD-4A, and HA-hRIPK1-WT (**b**) as indicated for 18 h. The cells lysates were immunoprecipitated with anti-Myc conjugated agarose beads and analyzed by western blotting using indicated antibody. Uncropped blots in the Source Data file. **c, d** MEF *Ripk1*$^{-/-}$ cells were reconstituted with 2xFV-mRIPK1-WT or 2xFV-mRIPK1-K612R by PMSCV retrovirus infection. The cells were pretreated with Nec-1s (10 μM), GSK'872 (10 μM) or vehicle for 30 min, and then treated with AP20187 (30 nM) in the presence of Z-VAD (25 μM) for indicated time. Cell survival was measured by Cell TiterGlo (**c**). Cell lysates were analyzed by western blotting using indicated antibodies (**d**). AP20187 (30 nM) was added to induce dimerization. Uncropped blots in the Source Data file. **e** Immunoprecipitation and western blotting analysis of DD interaction in HEK293T cells cotransfected expression plasmids of myc-hRIPK1-WT, myc-hRIPK1-K627R, myc-hRIPK1-ΔDD with flag-hTNFR1-DD, flag-hTRADD-DD, or flag-hFADD-DD as indicated. Cells lysates were immunoprecipitated with Flag M2 Agarose Affinity gel and analyzed by western blotting using indicated antibodies. Uncropped blots in the Source Data file. **f, g** RIPK1$^{+/+}$ and RIPK1$^{K612R}$ MEFs were reconstituted with 2xFV-RIPK3 by PMSCV retrovirus infection. Cells were pretreated with Nec-1s (10 μM), GSK'872 (10 μM) or vehicle for 30 min, and then treated with AP20187 (30 nM) in the presence of Z-VAD (25 μM) for indicated time to induce cell death. Cell survival was measured by Cell TiterGlo (**f**). Data are presented as mean ± SEM of *n* = 3 biologically independent samples. Two-way ANOVA with Bonferroni's multiple comparison test. The cells were lysed in SDS reducing sample buffer and analyzed by western blotting using indicated antibodies (**g**). Uncropped blots in the Source Data file. **h, i** FADD was deleted in MEF cells stably expressing 2xFV-RIPK3 by means of CRISPR/Cas9. The cells were treated with AP20187 (30 nM) in the presence of Z-VAD (25 μM) for indicated time. Cell survival was measured by CellTiterGlo (**h**). Data are presented as mean ± SEM of *n* = 3 biologically independent samples. Two-way ANOVA with Bonferroni's multiple comparison test. Cells were lysed in SDS reducing sample buffer and analyzed by western blotting using indicated antibodies (**i**). Uncropped blots in the Source Data file.

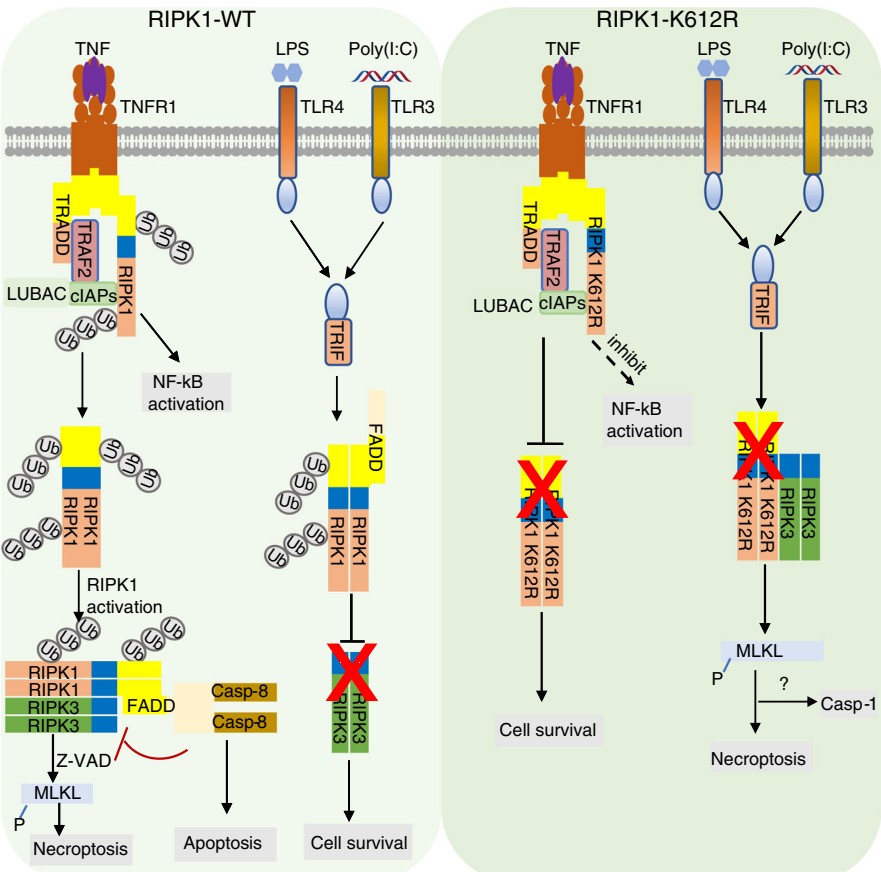

**Fig. 8 A model for RIPK1 K612 ubiquitination in TNFR1 and TLR3/4 signalings.** In TNFα-stimulated WT cells, RIPK1 is rapidly recruited to TNFR1 by direct binding to the death domain (DD) of TNFR1 or that of TRADD to initiate the formation of complex I. In complex I the ubiquitination of RIPK1 is modulated by multiple E3 ubiquitin ligases such as cIAP1/2 and LUBAC complex. Ubiquitination of RIPK1 mediates NF-κB activation. Ubiquitination of RIPK1 K612 modulates overall patterns of RIPK1 ubiquitination and promotes RIPK1-RIPK1 and RIPK1-TNFR1 interaction mediated by its DD, which leads to its activation and interaction with FADD/Caspase-8 to mediate RDA or with RIPK3 to mediate necroptosis when caspase-8 is inactivated. K612R mutation reduces the recruitment of RIPK1 and inhibits the activation of RIPK1 upon stimulation of TNFR1 by TNFα. RIPK1 K612R mutation inhibits cell death induced by TNFα through disrupting RIPK1 dimerization and RIPK1-TNFR1 interaction mediated by DD domain. In LPS or Poly (I:C)-stimulated WT cells, RIPK1 and RIPK3 interact with TRIF which regulates the activation of RIPK1 and RIPK3. Ubiquitination of RIPK1 K612 promotes RIPK1 ubiquitination, dimerization and interaction of RIPK1 with FADD, which in turn recruits FADD to suppress necroptosis and caspase-1 activation by restricting RIPK3 activation in response to TLR3 and TLR4. RIPK1 K612R mutation disrupts the interaction of RIPK1 and FADD to promote RIPK3-dependent necroptosis and inflammation induced by LPS or Poly(I:C).

activation of RIPK1 and cell death (Fig. 8). Thus, the RIPK1-DD may serve a dominant function over its ubiquitination in TNFR1 signaling as disrupting the recruitment of RIPK1 by blocking its DD-mediated interaction with TNFR1 is sufficient to inhibit its activation. In TLRs signaling pathway, however, the interaction of RIPK1 with FADD may be important to promote cell survival as inhibiting RIPK1 heterodimerization with FADD by K612R RIPK1 mutation sensitizes cells to RIPK3 activation in necroptosis, caspase-1 activation, and inflammation in response to TLR4 and TLR3 activation (Fig. 8). In addition, the ubiquitination of RIPK1 may serve a dominant function in regulating its activation in TLRs signaling. Reducing the overall ubiquitination of RIPK1 by R612R mutation promotes its binding with RIPK3 to mediate necroptosis and release of mature IL1β in TLRs signaling. Furthermore, RIPK3 kinase activity may provide a negative feedback in reducing RIPK1 activation as the activation of RIPK1 in RIPK1$^{K612R}$ mutant can be further enhanced by RIPK3 inhibitor GSK'872. Taken together, RIPK1-DD and ubiquitination of K612 in mRIPK1 and K627 in hRIPK1 preform opposing functions in regulating its activation in TNFR1 and TLR3/4 signaling pathways.

$Ripk1^{K612R/K612R}$ mice are normal at birth and viable as adult, but they spontaneously develop progressive intestinal inflammation and splenomegaly. Antibiotics treatment can totally rescue gut inflammation, indicating that commensal bacteria are crucial in triggering the gut inflammation. Studies show that regulators of either apoptosis, necroptosis or NF-κB can all contribute to regulation of gut inflammation, including intestinal epithelial cell (IEC)-specific RIPK1 knockout, caspase-8 or Fadd deletion or NEMO ablation[39,40,53,54]. Intestinal specific loss of RIPK1 ($Ripk1^{IEC-KO}$ mice), which develops severe intestinal inflammation which can be blocked by caspase-8, or Tnfr1 deficiency, or antibiotics treatment, but not by Ripk3 deficiency[39], demonstrating the dominant role of RIPK1 scaffold function by regulating Caspase-8/FADD activation in protecting intestinal health. $Fadd^{IEC-KO}$ mice develop spontaneous colitis which can be blocked by Ripk3, or Myd88 deficiency, or bacteria depletion[40], suggesting that the interaction of RIPK1 with FADD is important to block the activation of RIPK3 in intestine triggered by bacteria. The colonic inflammatory phenotype of $Ripk1^{K612R/K612R}$ mice also show some similarity to that of FADD$^{IEC-KO}$ or RIPK1$^{IEC-KO}$ mice, as RIPK1 K612R mutation blocks the binding with FADD and thus, our results suggest that RIPK1 is required to recruit FADD to suppress inflammation in response to TLRs signaling. Since K612R mutation leads to reduction of RIPK1 expression in multiple tissues in adult mice which can be rescued by antibiotic treatment, our results also suggest that ubiquitination of K612 RIPK1 is involved in regulating RIPK1 levels in response to TLRs signaling. Consistently, we demonstrate dramatic changes in the overall ubiquitination patterns in K612R mRIPK1 and K627R hRIPK1 compared to that of WT, suggesting K612/K627 in mRIPK1/hRIPK1 regulate the overall ubiquitination modification patterns. However, since $Ripk1^{K612R/K612R}$ mice show no loss of Paneth cells or goblet cells, unlike that of $Fadd^{IEC-KO}$ mice and $Ripk1^{IEC-KO}$ mice, ubiquitination of K612/K627 RIPK1 might primarily control the production of proinflammatory cytokines. Impaired NF-κB signaling in $Ripk1^{K612R/K612R}$ mice might also contribute to the inflammation by reducing the expression of key suppressors of RIPK1 and RIPK3, including cFLIP$_L$, cIAPs, and A20[13,16,55,56]. Thus, our study highlights the important role of RIPK1 mediated inflammatory response independent of cell death in intestine. $Ripk1^{K612R/K612R}$ mice may provide a good model for studying the mechanism by which reduction of RIPK1 expression in adults leads to inflammatory bowel disease (IBD) as Ripk1-deficient condition in humans.

While the activation of RIPK1 kinase promotes the activation of RIPK3 in TNFR1 signaling pathway, the scaffold of RIPK1 can inhibit the activation of RIPK3 in the cytosol[57]. RIP homotypic

interaction motif (RHIM) in RIPK1 has been shown to prevent the RHIM-containing adapter protein ZBP1 (Z-DNA binding protein 1; also known as DAI or DLM1) from activating RIPK3[58,59]. Our result suggests that RIPK1-DD is also involved in suppressing the activation of RIPK3 in TLRs signaling pathway, likely by engaging FADD. The activation of RIPK1 in Ripk1$^{K612R}$ BMDMs might be mediated by homotypic interaction of the RIPK1–RHIM with the RIPK3–RHIM, which is known to mediate the interaction of RIPK1 and RIPK3 in complex II during necroptosis. Activation of necroptosis in Ripk1$^{K612R}$ BMDMs with LPS stimulation can promote cell lysis which can promote the activation of caspase-1, cleavage of pro-IL1β, and subsequent release of mature IL1β. Our study suggests the possibility that activation of RIPK3 can provide a mechanism to mediate the release of cytokines such as mature IL1β, which lacks a signal peptide required for conventional secretion.

In conclusion, our study highlights the important role of RIPK1-DD and ubiquitination of K612 in mRIPK1 and K627 in hRIPK1 in maintaining intestine homeostasis. Even though K612R RIPK1 mutation protects against TNFR1 mediated RIPK1-dependent apoptosis and necroptosis, $Ripk1^{K612R/K612R}$ mice and cells are hypersensitive to TLRs signaling mediated inflammation involving caspase-1 activation and IL1β secretion. Thus, the scaffold function of RIPK1 mediated by DD not only suppresses apoptosis and necroptosis, but also caspase-1-mediated IL1β processing in response to TLRs signaling. Our study provides mechanistic insights into the cross-talks between multiple forms of cell death programs, including apoptosis, necroptosis, and caspase-1 activation.

## Methods

**Cell culture and cell line generation**. HEK293T cells, MEFs, and BMDMs were cultured in DMEM medium (Gibico) supplemented with 10% heat-inactivated fetal bovine serum (FBS), 100 unites/ml penicillin and streptomycin. Jurkat cells were cultured in RPMI-1640 medium (Gibico) supplemented with 10% heat-inactivated FBS, 100 unites/ml penicillin and streptomycin. All cells were cultured at 37 ℃ with 5% $CO_2$. Primary mouse embryonic fibroblast (MEFs) were obtained from E13.5 embryo, immortalized with SV40 large T antigen (SV40LT) and selected with 1 μg/ml puromycin; flag-RIPK1 (WT or mutant) MEFs were generated by stably reconstituting RIPK1$^{-/-}$ MEFs with retrovirus transduction of flag-RIPK1-WT or mutant constructs and selected with 1 μg/ml puromycin; primary bone marrow-derived macrophages (BMDMs) were obtained from mouse rear leg's bone marrow and differentiated into macrophages for 6 days with 20% filtered L929 conditional medium (a source of macrophage colony-stimulating factor).

**Reagents and antibodies**. The sources of the materials used in this study: human recombinant soluble TNFα (Novoprotein Scientific), mouse recombinant soluble TNFα (R&D Systems), flag mouse TNFα (homemade), 5Z-7-Oxozeaenol (Sigma-Aldrich), GSK'872 (Calbiochem), Z-VAD.fmk (Selleckchem), R-7-Cl-O-Nec-1 (Nec-1s), and SM-164 (custom synthesized). Antibodies from Cell Signaling Technology: RIPK1 (3493), A20 (5630), Caspase-8 (4927), cl-Caspase-8 (8592), Caspase-3 (9662), cl-Caspase-3 (9661), p44/42 MAPK (4695), P-p44/42 MAPK (4370), p38 (9212), P-p38 (9216), JNK (9252), P-JNK (4671) P-IKBα (2859), IKBα (4814), P-IKKα/β (2078), IKKα (2682), IKKβ (2370), and mouse P-RIPK3 (91702). Antibodies from Abcam: mouse P-S345-MLKL (ab196436) and FADD (ab124812). Antibodies from Santa Cruz: mouse FADD (SC-6036), TRAF2 (SC-876), TNFRSF1A (SC-8436), TRADD (SC-7868), RBCK1 (SC-365523), GFP (SC8334), IKKγ (SC-8330), and CYLD (SC-74435). Antibodies from Proteintech: SHARPIN (14626-1-AP) and actin (66009). Caspase-1 p20 (AG-20B-0042) from Adipogen Life Science. IL-1β (GTX74034) from Gentex. α-Tubulin (T9026) from Sigma-Aldrich. HRP conjugated goat anti rabbit IgG (H + L, 31460) and HRP conjugated goat anti mouse IgG (H + L, 31430) (Thermo Fisher Scientific). Goat anti-rat IgG H&L (ab214882) and goat anti-rabbit IgG H&L (ab214880) from Abcam. Anti-pS166-mRIPK1 and anti-mRIPK1 rabbit monoclonal abs (Biolynx). Anti-mRIPK3, anti-mMLKL, anti-m-cIAP1, anti-mHOIL and anti-mHOIP (homemade). Murine TNFRSF1A (R&D Systems). Flag M2 Agarose Affinity gel (A2220) (Sigma-Aldrich). K63 and linear ubiquitin chain abs were kindly provided by Dr. Vishva M Dixit of Genentech.

**Mice**. $Ripk1^{K612R/K612R}$ knockin mice in B6D2F1 background were generated by mutating lysine codon (AAA) at position 612 to arginine (CGC) using CRISPR/Cas9 (Supplementary Fig. 2). Genotyping of K612R mice was conducted with mouse-tail DNA by PCR and DNA product sequencing. $Ripk1^{K612R/K612R}$ knockin

mice were backcrossed with C57BL/6 for eight generations. $Ripk3^{-/-}$ mice were kindly provided by Dr. Xiaodong Wang of National Institute of Biological Science, Beijing. All animals were maintained in a specific pathogen-free environment, and animal experiments were conducted according to the protocols approved by the Standing Animal Care Committee at the Interdisciplinary Research Center of Biology and Chemistry, Shanghai Institute of Organic Chemistry. The license registration number of experimental animal management is SYXK (Shanghai) 2016-0001. The environmental temperature is $23 \pm 2\,°C$ and the humidity is $55 \pm 5\%$. The mice were free to food intake and water with 12 h interval of illumination and darkness. The cages and litter were regularly replaced. Primers for genotyping were listed in Supplementary Table 3.

**Depletion of commensal bacteria**. Littermates of WT and $Ripk1^{K612R}$ mice (3–4 weeks old) were treated with 200 mg ciprofloxacin, 1 g ampicillin, 1 g metronidazole, and 500 mg vancomycin per liter in drinking water. Drinking water and bedding were refreshed every 3 days[39]. After 4 weeks, mice were sacrificed, and tissues were collected for qRT-PCR analysis, western blotting, or histology.

**Co-immunoprecipitation and immunoblot analysis**. Cell lysates were prepared in following lysis buffer: 50 mM Tris-HCl (pH 7.5), 150 mM NaCl, 1 mM EDTA, 1% NP40, 10% glycerol supplemented with 1 mM PMSF, 1× protease inhibitor cocktail (Roche), 10 mM β-glycerophosphate, 5 mM NaF, and 1 mM Na3VO4. Cells were lysed on ice for 30 min and centrifuged at $20,000 \times g$ for 15 min at 4 °C. The cell lysates were incubated with indicated antibody overnight at 4 °C and immuno-complex was captured by protein A/G agarose (Life Technology). After extensive washes, beads were boiled in SDS reducing sample buffer and eluted products were separated by SDS PAGE which was transferred to PVDF membrane (Millipore) and analyzed with indicated antibodies.

**Immunoprecipitation of endogenous RIPK1 for quantitative mass spec analysis**. Fifteen 150 mm-plates of MEF cells at 90% confluency per sample were treated or without 100 ng/ml mouse TNF for 5 min. The cells were lyzed in 2 ml RIPA buffer [50 mM Tris-HCl pH 7.5, 150 mM NaCl, 1 mM EDTA, 1% NP40, 0.1% SDS, 10% glycerol supplemented with 1 mM PMSF, 1× protease inhibitor cocktail (Roche), 10 mM β-glycerophosphate, 5 mM NaF, and 1 mM Na3VO4] and incubated on ice for 30 min. The cell lysates were quantified with BCA. Seventy milligram total protein lysate was incubated with RIPK1 antibody (Biolynx, clone: YJY-4-3) overnight at 4 °C and immunocomplex was captured by protein A/G agarose. Immunoprecipitates were washed with RIPA buffer for five times, and with ddH2O for four times. After washes, the beads were processed for mass spectrometry analysis.

**Mass spectrometry**. Immunoprecipitated RIPK1 was trypsin-digested on beads. The resulting peptides were subjected to enrichment of diGly peptides using antibody against ubiquitin remnant motif (K-ε-GG) (PTM biolabs, lnc). The enriched diGly peptides were analyzed on the Q Exactive HF-X mass spectrometer, and the data were acquired using xCalibur3.1 from Thermo Fisher Scientific. DiGly peptides were identified and quantified using MaxQuant[60]. The tandem mass spectra were searched against UniProt mouse protein database together with a set of commonly observed contaminants. The precursor mass tolerance was set as 20 ppm, and the fragment mass tolerance was set as 0.1 Da. The cysteine carbami-domethylation was set as a static modification, and the methionine oxidation as well as lysine with a diGly remnant were set as variable modifications. The FDR at peptide spectrum match levels were controlled below 1%.

**Cell viability assay**. Cell survival was assessed using ATP luminescence assay CellTiterGlo (Promega).

**Gene expression analysis**. The intestinal section tract was removed and separated into ileum, caecum, and colon. Total RNA was isolated from MEFs and intestinal section using TRIzol Reagent (Invitrogen). RNA concentration was measured using the Nanodrop ND-1000 spectrophotometer (Thermo scientific). cDNA was prepared with M-MLV reverse transcriptase (TaKaRa). cDNA(10 ng) of each sample was used for quantitative PCR with SYBR Green PCR Master Mix (Biotool). Data were analyzed according to the CT method. The primers for murine inflammatory cytokines were listed in Supplementary Table 3.

**Gene deletion in MEF cells**. MEF cells were infected with lentiviruses carrying either LentiCRISPR v2 or LentiCRISPR v2-sgTRADD or LentiCRISPR v2-sgFADD. After 48 h, the cells were selected by 5 μg/ml puromycin for 3 days. The sgRNAs for Fadd and Tradd were listed in Supplementary Table 3.

**Quantification of cytokines**. Cytokines from sera or cell supernatants were quantified by R&D DuoSet ELISA kit, including TNFα (DY410), IL-6 (DY406), and IL-1β (DY401). Cytokine ELISA was performed according to manufacturer's instructions.

**Gut sample preparation for western blotting**. Colon (~1 cm) and ileum (~1 cm) from adult mice were collected and lysed in 1 ml 1% NP40 plus 0.1% SDS with TissueLyser II(QIAGEN) for eight times. The lysates were then centrifuged at $20,000 \times g$ for 10 min and supernatant was collected as soluble fraction. The pellets were washed once with RIPA lysis buffer, then lysed in loading buffer with 4% SDS (without DTT and bromophenol blue) for 24 h at 16 °C, boiled at 100 °C for 10 min, and centrifuged at $20,000 \times g$ for 10 min. The supernatant was collected as insoluble fraction and lysed in loading buffer for western blotting.

**Fluorescence-activated cell sorting (FCAS)**. Total cell suspensions were pre-pared from lymph nodes, thymus, spleen and bone marrows, and red blood cells were lysed by incubation with ACK lysis. Single cell suspensions were pre-incubated with purified rat anti-mouse CD16/CD32 (Biolegend, 101302) to block FcγR, and then stained with fluorochrome conjugated cell surface antibodies: CD45 (BD Bioscience, 553081), CD11b (Biolegend, 101212), Gr-1 (Biolegend, 108406), CD3 (Biolegend, 100221), B220 (Biolegend, 103212), CD4 (Biolegend, 100540), and CD8 (Biolegend, 100706). Data from immunofluorescent staining of cells were acquired on Guava easyCyte HT (Millipore) and analyzed using guavasoft 3.1.1.1. The gating strategy for FCAS is shown in Supplementary Fig. 9.

**Histology**. Intestine tissues were fixed in 4% paraformaldehyde, embedded in par-affin, and cut into 5 μm sections and dewaxed, rehydrated, and then stained with hematoxylin & eosin or AB/PAS. For immunohistochemistry of Gr-1 and F4/80, cryostat sections (10 μm thick) were blocked with 3% H2O2 and then with 5% goat serum in PBST (PBS with 0.1% Triton X 100). For immunohistochemistry of Ki67 and lysozyme, paraffin-embedded sections (8 μm thick) were dewaxed and rehy-drated, before incubated in 0.01 M sodium citrate buffer (pH 6.0) for heat-induced antigen retrieval. The sections were blocked with 3% H2O2 and then with 5% goat serum in PBST (PBS with 0.1% Triton X 100) and incubated with primary antibodies at 4 °C overnight and then washed 3 times with PBST before incubating with sec-ondary antibodies. The signals were detected by SignalStain® DAB Substrate kit (CST, 8059). Primary antibodies for immunohistochemistry used: anti-Ki67 (CST, 12202), anti-F4/80 (Biolegend, 123102), anti-Gr-1 (BD, 550291) and lysozyme (Invitrogen, PA5-16668). Secondary antibodies for immunohistochemistry used: goat anti-rat IgG H&L (Abcam, ab214882), goat anti-rabbit IgG H&L (Abcam, ab214880).

**Statistical analysis and reproducibility**. The cell data are presented as mean ± SEM (standard error of mean) of triplicate wells from one representative experi-ment. All immunoblots from cell samples were repeated at least three times independently with similar results. The mouse data are presented as mean ± SEM of indicated $n$ values. Immunoblots from mouse samples were repeated once with similar results. The statistical comparisons were performed using an unpaired two-tailed Student's $t$-test between two groups, or one-way ANOVA with Dunnett's multiple comparison test among multiple groups with a single control, or two-way ANOVA with Bonferroni's multiple comparison test among different groups. Graphpad Prism software version 7.0 was used for data analysis. $*P \le 0.05$; $**P \le 0.01$; $***P \le 0.001$; $****P \le 0.0001$; NS not significant.

**Reporting summary**. Further information on research design is available in the Nature Research Reporting Summary linked to this article.

## Data availability
The mass spectrometry data have been deposited at iProX under the accession number PXD022157. Mouse protein database were obtained from Uniprot (https://www.uniprot.org/). All data are available in Source Data File. Uncropped western blot scans are shown in the Source Data File.

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

## Acknowledgements

The authors thank Dr. Vishva Dixit of Genentech for M1 and K63 ubiquitin chain abs. This work was supported by the National Key R&D Program of China (2016YFA0501900), the China National Natural Science Foundation (31530041, 21837004 and 91849204), the Chinese Academy of Sciences (XDB39030000 and XDB39030600), China National Natural Science Youth Foundation (31701210) and the Science and Technology Commission of Shanghai Municipality (18JC1420500).

## Author contributions

J.Y. and B.S. designed and directed the experiments; X.L. designed and conducted majority of the experiments; X.H., W.L., G.L., X.J.L., M.Z., and H.Z. conducted specific experiments; Y.L. and H.P. generated antibodies for mRIPK1 pS166 and anti-mRIPK1; L.S. and L.Q. provided expertise of generating RIPK1 K612R knockin mice.

## Competing interests

J. Yuan is a consultant for Denali Therapeutics and Sanofi. The other authors declare no competing interests.
