## [Peer Review File · Nature Communications]

REVIEWER COMMENTS

Reviewer #1 (Remarks to the Author):

The study of Li and colleagues attempts to address the role of ubiquitin (Ub) modification of mLys612/hLys627 in RIPK1. They show that a K612R mutation in the death domain (DD) of mouse RIPK1 interferes with recruitment of the kinase into TNFR1 complex I in MEFs (Fig. 2). Accordingly, RIPK1-K612R knock-in (KI) MEFs and BMDMs are resistant to forms of TNF-induced cell death that require the kinase activity of RIPK1 (Fig. 1).

Because the authors (and others before them) can detect Ub on ectopic wild-type (WT) RIPK1 at K612/K627, they infer from their results that Ub of RIPK1 K612 is important for TNF signaling. However, an important question remains: where does endogenous RIPK1 get the K612-Ub modification? In the cytoplasm or only in complex I? The answer gets to whether impaired Ub of cytoplasmic RIPK1 prevents RIPK1 recruitment to complex I. If only RIPK1 in complex I undergoes Ub at K612, then the effect of the K612R mutation on recruitment to complex I is probably unrelated to Ub at K612 and may just be due to improper folding of the DD.

The authors then emphasize the notion that Ub at K612/K627 in RIPK1 is a "master regulator" of Ub modification of other RIPK1 residues. However, supporting evidence relies on overexpression of WT vs K627R RIPK1 in 293T cells (Fig. 3c), where the physiological relevance of the Ub sites observed is far from clear. Is endogenous WT RIPK1 and RIPK1-K612R modified in the same way? Moreover, is differential Ub increased/reduced/the same upon TNF stimulation? Commercial antibodies to RIPK1 are excellent and well suited to this type of IP/MS experiment.

Next the authors report that K612R KI mice fail to thrive and develop gut inflammation and splenomegaly that is ameliorated by antibiotics or RIPK3 loss (Fig. 4, 5 & S7). These data confirm the importance of the K612 residue in RIPK1 for restricting microbiome- and RIPK3-dependent inflammation. However, whether this phenotype reflects impaired Ub of the RIPK1 DD, as claimed, versus a RIPK1 hypomorph with an impaired DD structure remains unclear. Failure of the RIPK1 DD to recruit FADD/casp8, presumably blocks suppression of RIPK1/RIPK3 signaling by casp8, and hence explains the increased sensitivity of KI BMDMs to TLR3 and TLR4 toxicity (Fig. 6).

In sum, many observations of the authors require further fleshing out to justify the conclusions drawn.

Other issues:

- Introduction (page 3): Li et al 2019 PNAS 116:970-975 should be cited in addition to ref. 5 when referencing RIPK1-deficient humans.
- Fig. S1c: why does Nec-1s not protect against death induced by TS- or T5Z-7 in cells expressing WT RIPK1? The main text suggests this should be RDA and therefore inhibited by Nec-1s.
- Page 7, line 4: it is more accurate to say "reduced" rather than "inhibited" when describing phosphorylation of RIPK1, RIPK3 and MLKL in Fig. 1d.
- Fig. 1e: WBs for pRIPK1-S166 are very poor quality. Authors should IP total RIPK1, and then WB pRIPK1-S166 to get a cleaner result. However, essentially the same data are shown in Fig. 2a and 2b (minus cleaved casp3) and the p-RIPK1 WBs in fig. 2 are a lot cleaner. Therefore, they should just remove Fig. 1e. Indeed, the data in Fig. 1d should also be removed because they are redundant with those shown in Fig. 2c and 2d. The text describing these figures could be condensed accordingly.
- Fig. 2a: if the upper band in all lanes of the cleaved casp8 WB after FADD IP is the heavy chain of the precipitating antibody, then it should be labelled as such. They appear to have used far too much of the IP antibody because this band isn't nearly as prominent in Fig. 2b. A similar label appears

warranted for the lower band in the IP RIPK3/WB RIPK3 panel of Fig. 2c.

- Fig. 2e: it is very hard to see bands in the IP Flag/WB HOIL panel. Can the authors develop it longer so that you can see the purported bands clearly?

- The y-axis of graphs indicating % survival should top out at 100%, not 150%. This adjustment will help convince readers of the seemingly small differences between WT and KI cells, particularly in the likes of Fig. S3b.

- Fig. S4a: I agree that p-ERK, p-JNK, and p-IKK are reduced, but I struggle to see the claimed reduction in p-IkB or impaired degradation of IkB in the TNF-treated KI cells. The description on page 9 of the main text should be amended.

- Fig. 3a: this IP/WB experiment addressing the Ub linkages on RIPK1 is flawed. Based on the methods, they did not perform IPs on SDS-boiled/denatured lysates. Therefore, the ubiquitin chains detected could be on RIPK1-associated proteins, rather than RIPK1 itself.

- Fig. 3d is presented as supporting evidence of K612 Ub being a master regulator of RIPK1 Ub, but this really is a stretch. All forms of TNF-induced Ub on RIPK1 are reduced because the recruitment of RIPK1 is impaired, and as discussed above, the authors have failed to prove that this recruitment defect stems from impaired Ub.

- Fig. 4b: if males and females are used in this graph, then they should be presented separately because the average male body weight at 8 wks is greater than the average female body weight. If a single sex is presented then this should be indicated in the legend.

- Fig. 6d, h and text on page 16: absent any genetic rescue with MLKL loss, the authors cannot conclude that "necroptosis" is driving LPS-induced IL-1b release from KI BMDMs. RIPK3 has been shown to have non-necroptotic functions and thus its loss, while suggestive, is insufficient to prove necroptosis involvement.

Reviewer #2 (Remarks to the Author):

The manuscript by Li et al analyzed different ubiquitination sites in RIP1 for their function in cell death. K627 in human RIP1 was identified as a ubiquitination site that controls the overall ubiquitination of RIP1's kinase domain and DD domain mediated interaction with other proteins. K627R mutation inhibits TNF-induced cell death. The authors also generated RIP1 K612R mouse. In vitro and in vivo data showed that K612R mutation led to increased RIP3 dependent necroptosis, which sensitized cells to inflammasome activation by TLR3/4. The study was well designed and conducted. I have a few minor questions.

1. Since caspase-1 activation and IL-1beta secretion occurred when K612 was mutated to R, a brief introduction and discussion of inflammasome is needed.
2. In K612R cells, complex I and II reduced, but necrosome increased. It is not very clear to me whether and how the impaired complex I formation by K612R mutation affected the formation of complex II and necrosome? Maybe a brief discussion is needed.
3. Page 13. The sentence "...lead to defective NF-kB pathway an a strong inflammatory response....." might cause confusion that defective NF-kB enhanced inflammation. It would be better to reword the sentence.
4. Figure 6d. K62R/K612R needs to be corrected.
5. The data in the manuscript seem to suggest that K612R did not simply mimic un-ubiquitinated RIP1, but also gained additional function(s). The authors need to discuss this issue.

Reviewer #3 (Remarks to the Author):

In this manuscript, Li et al. present convincing evidence that K612 in mRIPK1/K627 in hRIPK1 plays a critical role in regulating TNFR1 and TLR signaling as a master ubiquitination site that controls the ubiquitination pattern of RIPK1. They generated Ripk1K612R/K612R knockin mutant mice to extensively dissect the function of K612 in vitro and in vivo. Interestingly, they provided evidence that K612R mutation inhibits TNF α induced-RDA and necroptosis, but promotes necroptosis and caspase-1 activation stimulated by TLR3/4. Ripk1K612R/K612R mice develop adult-onset intestinal inflammation and splenomegaly, which can be rescued by antibiotic treatment or co-ablation of Ripk3. Further investigation showed that RIPK1 K612 is important for DD mediated interaction of RIPK1 with different DD-containing proteins, which may explained the distinct function of K612 in TNFR1 and TLRs signaling. The data shown are of very good quality. Here are several aspects that need improvement.

1. In Figure 3b, M1 ubiquitination of hRIPK1 K627R is shown to be comparable with that of WT, but M1 and K63 ubiquitinated RIPK1 in K612R mutations cells were reduced compared to that of WT cells upon TNF α /SM-164/zVAD treatment. This difference should be discussed.

2. In Figure 4, authors only provided results of macrophages and neutrophil in colon. Other type immune cells should also be examined in thymus, spleen and lymph nodes to provide a complete picture of immune homeostasis in Ripk1K612R/K612R mice. Was there accompanied with lymphadenopathy in Ripk1K612R/K612R mice?

3. In Figure 4c, RIPK1 protein level in multiple tissue from aged Ripk1K612R/K612R mice was greatly decreased compared to that from WT mice. Authors should examine if RIPK1 K612R has moved into the insoluble fraction.

4. In addition, whether K612 in mRIPK1/K627 in hRIPK1 controlling the ubiquitination pattern on other sites of RIPK1 is mediated by RIPK1 kinase activity should be test or at least discussed in text.

Reply to reviewer 1

Reviewer #1 (Remarks to the Author):

The study of Li and colleagues attempts to address the role of ubiquitin (Ub) modification of mLys612/hLys627 in RIPK1. They show that a K612R mutation in the death domain (DD) of mouse RIPK1 interferes with recruitment of the kinase into TNFR1 complex I in MEFs (Fig. 2). Accordingly, RIPK1-K612R knock-in (KI) MEFs and BMDMs are resistant to forms of TNF-induced cell death that require the kinase activity of RIPK1 (Fig. 1).

Because the authors (and others before them) can detect Ub on ectopic wild-type (WT) RIPK1 at K612/K627, they infer from their results that Ub of RIPK1 K612 is important for TNF signaling. However, an important question remains: where does endogenous RIPK1 get the K612-Ub modification? In the cytoplasm or only in complex I? The answer gets to whether impaired Ub of cytoplasmic RIPK1 prevents RIPK1 recruitment to complex I. If only RIPK1 in complex I undergoes Ub at K612, then the effect of the K612R mutation on recruitment to complex I is probably unrelated to Ub at K612 and may just be due to improper folding of the DD.

The authors then emphasize the notion that Ub at K612/K627 in RIPK1 is a “master regulator” of Ub modification of other RIPK1 residues. However, supporting evidence relies on overexpression of WT vs K627R RIPK1 in 293T cells (Fig. 3c), where the physiological relevance of the Ub sites observed is far from clear. Is endogenous WT RIPK1 and RIPK1-K612R modified in the same way? Moreover, is differential Ub increased/reduced/the same upon TNF stimulation? Commercial antibodies to RIPK1 are excellent and well suited to this type of IP/MS experiment.

Reply: We thank the reviewer for the constructive comments. To address whether K612 is involved in regulating the ubiquitination of endogenous RIPK1, we immunoprecipitated endogenous RIPK1 from 15 x 150mm plates of RIPK1-WT-MEFs and RIPK1-K612R knockin MEFs that were treated with or without TNF α for 5 mins using a validated rabbit monoclonal anti-RIPK1 antibody and analyzed the ubiquitination pattern of endogenous RIPK1 by quantitative mass spectrometry. In control WT MEFs without TNF stimulation, many lysine residues in RIPK1 were highly ubiquitinated, including 11 lysine residues in the kinase domain (K20, K30, K45, K65, K115, K137, K140, K153, K163, K167 and K307), 4 lysine residues in the intermediate domain (K376, K392, K395, and K429) and 4 lysine residues in the DD (K589, K612, K627 and K633). Similar to the mass spec data on the ubiquitination changes of K612R RIPK1 expressed in 293T cells, the majority of these sites in the kinase domain and intermediate domain of endogenous K612R RIPK1 showed a reduction in ubiquitination levels (Fig. 3e-f). The ubiquitination levels of 15 lysine residues (out of the 19 ubiquitinated lysine residues in WT-RIPK1) on K612R-RIPK1 in K612R-RIPK1 knockin MEFs under control condition were reduced compared to that of WT RIPK1 in WT MEFs, including K20, K30, K45, K46, K115, K137, K140, K153, K163, K167, K307, K392, K395, K429 and K627 (Fig. 3f and Supplementary Fig. 5b). The ubiquitination levels of K65 and K376 on K612R-RIPK1 in RIPK1-K612R knockin MEFs under control conditions were comparable to that of WT-

RIPK1 in WT MEFs; while ubiquitination levels of 3 lysine residues in the DD of K612R-RIPK1 (K589, K619 and K633) and one lysine residue in the kinase domain (K105) in K612R-RIPK1 knockin MEFs were increased compared to that of WT-RIPK1, similar to that K612R RIPK1 expressed in 293T cells (Fig. 3f, supplementary Fig.5b).

We further compared the patterns of ubiquitination in endogenous WT RIPK1 and K612R RIPK1 in WT and RIPK1-K612R knockin MEFs stimulated by TNF α for 5 min. In WT MEFs stimulated by TNF α for 5 min, the ubiquitination levels of 19 lysine residues, including K20, K45, K46, K65, K105, K137, K140, K153, K163, K167, K376, K392, K395, K429, K550, K589, K612, K619 and K633, were increased; the ubiquitination levels of 4 lysine residues, including K30, K115, K307 and K627, were decreased (Fig 3f; Supplementary Fig. 5b). In K612R-RIPK1 knockin MEFs stimulated by TNF α for 5 min, the ubiquitination levels of 17 lysine residues in K612R-RIPK1 were reduced compared to that of RIPK1 in WT MEFs, including K20, K30, K45, K46, K65, K105, K137, K140, K153, K163, K167, K376, K392, K395, K429, K550, K619; while the ubiquitination levels of K589 were comparable to that of WT; and the ubiquitination levels of 3 lysine residues in K612R-RIPK1 in K612R-RIPK1 knockin MEFs, including K115, K307 and K633, were increased compared to that of WT. Thus, while TNF α stimulation led to a rapid increase in the ubiquitination of multiple sites all over RIPK1, the ubiquitinations of many of these lysine residues were blocked or reduced by K612R mutation (Fig 3f; Supplementary Fig. 5b). These data suggest that K612R mutation can affect the overall patterns of RIPK1 ubiquitination in the cytoplasm under unstimulated condition as well as in complex I after TNF α stimulation for 5 min.

These new data have been incorporated into revised manuscript as Fig. 3f and Supplementary Fig. 5b. We also tuned down the conclusion to call K612 in mRIPK1/K627 in hRIPK1 as a “key ubiquitinating site that regulates the overall ubiquitination pattern of RIPK1”, instead of “a master ubiquitination site that controls the overall ubiquitination of RIPK1”.

Next the authors report that K612R KI mice fail to thrive and develop gut inflammation and splenomegaly that is ameliorated by antibiotics or RIPK3 loss (Fig. 4, 5 & S7). These data confirm the importance of the K612 residue in RIPK1 for restricting microbiome- and RIPK3-dependent inflammation. However, whether this phenotype reflects impaired Ub of the RIPK1 DD, as claimed, versus a RIPK1 hypomorph with an impaired DD structure remains unclear. Failure of the RIPK1 DD to recruit FADD/casp8, presumably blocks suppression of RIPK1/RIPK3 signaling by casp8, and hence explains the increased sensitivity of KI BMDMs to TLR3 and TLR4 toxicity (Fig. 6).

In sum, many observations of the authors require further fleshing out to justify the conclusions drawn.

Reply:

Our data showed that RIPK1 levels in newborn RIPK1-K612R mice were normal, comparable to that of RIPK1-WT tissues (Fig.4c,d). The reduction of RIPK1 levels in RIPK1-K612R mice were progressive: first detectable at 3 weeks of age and more severe at 6 weeks of age. The progressive loss of RIPK1 level could be rescued by antibiotic treatment (Fig. 5e). These data suggest that the reduction in the levels of K612-RIPK1 in the *Ripk1*^{K612R/K612R} knockin mice is induced by intestinal bacteria-mediated inflammation, rather than a constitutive hypomorph of

RIPK1. Consistently, RIPK1^{+/-} mice do not develop gut inflammation. These data support the importance of the K612 residue in RIPK1 in restricting microbiome- and RIPK3-dependent inflammation.

Other issues:

- Introduction (page 3): Li et al 2019 PNAS 116:970-975 should be cited in addition to ref. 5 when referencing RIPK1-deficient humans.

Reply: We have added “Li et al 2019 PNAS 116:970-975” as ref.4.

- Fig. S1c: why does Nec-1s not protect against death induced by TS- or T5Z-7 in cells expressing WT RIPK1? The main text suggests this should be RDA and therefore inhibited by Nec-1s.

Reply: RDA is most effective in murine cells, but not so effective in many human cell lines. Jurkat cells are not very sensitive to RDA induced by T/S or T5z-7, which is why Nec-1s could not protect TS- or T5Z-7 induced necroptosis. Thus, we have removed TS- and T5Z-7 from Supplementary Fig. 1d to avoid confusion.

- Page 7, line 4: it is more accurate to say “reduced” rather than “inhibited” when describing phosphorylation of RIPK1, RIPK3 and MLKL in Fig. 1d.

Reply: Fig. 1d has been removed per advice of this reviewer. In the corresponding Fig. 2c and 2d, we changed “inhibited” to “reduced” as advised by this reviewer. We also removed “strongly” in Fig. 2a and 2b.

- Fig. 1e: WBs for pRIPK1-S166 are very poor quality. Authors should IP total RIPK1, and then WB pRIPK1-S166 to get a cleaner result. However, essentially the same data are shown in Fig. 2a and 2b (minus cleaved casp3) and the p-RIPK1 WBs in fig. 2 are a lot cleaner. Therefore, they should just remove Fig. 1e. Indeed, the data in Fig. 1d should also be removed because they are redundant with those shown in Fig. 2c and 2d. The text describing these figures could be condensed accordingly.

Reply: We thank the reviewer for this suggestion. We agree that the original Fig. 1d/e and Fig. 2a/b were redundant. We removed Fig. 1d,1e, and added caspase3-fl and caspase3-cleavage in Fig. 2a and 2b.

- Fig. 2a: if the upper band in all lanes of the cleaved casp8 WB after FADD IP is the heavy chain of the precipitating antibody, then it should be labelled as such. They appear to have used far too much of the IP antibody because this band isn't nearly as prominent in Fig. 2b. A similar label appears warranted for the lower band in the IP RIPK3/WB RIPK3 panel of Fig. 2c.

Reply: The upper band in all lanes of the cleaved casp8 WB after FADD IP is the heavy chain of IgG as this band did not exist in caspase-8 western (second western blot). * is now put to label the IgG in both Fig. 2a and 2b.

Fig. 2c, the IP/western RIPK3 does not have the same problem, as we used two different RIPK3 abs for this experiment: a mouse RIPK3 ab was used for IP and a rabbit RIPK3 ab was used in western. The lower band in the IP RIPK3/WB RIPK3 panel is total RIPK3, the upper band is phosphorylated RIPK3 triggered by necroptosis stimuli.

- Fig. 2e: it is very hard to see bands in the IP Flag/WB HOIL panel. Can the authors develop it longer so that you can see the purported bands clearly?

Reply: We repeated this experiment and obtained data with improved quality. HOIL runs very close to the heavy chain of IgG on SDS page (Fig. 2e).

- The y-axis of graphs indicating % survival should top out at 100%, not 150%. This adjustment will help convince readers of the seemingly small differences between WT and KI cells, particularly in the likes of Fig. S3b.

Reply: We adjusted the y-axis to 100% in Fig. S3a,b.

- Fig. S4a: I agree that p-ERK, p-JNK, and p-IKK are reduced, but I struggle to see the claimed reduction in p-IkB or impaired degradation of IkB in the TNF-treated KI cells. The description on page 9 of the main text should be amended.

Reply: Yes, we agree with this point, although p-IKK α/β is decreased in TNF α stimulated RIPK1-K612R cells, p-IkB α and degradation of IkB α were comparable between RIPK1-K612R and RIPK1-WT cells. It is possible that the amount of activated IKK α/β in RIPK1-K612R is enough to phosphorylate IkB α . We have revised text to say: However, p-IkB α and degradation of IkB α were comparable between RIPK1-K612R and RIPK1-WT cells.

- Fig. 3a: this IP/WB experiment addressing the Ub linkages on RIPK1 is flawed. Based on the methods, they did not perform IPs on SDS-boiled/denatured lysates. Therefore, the ubiquitin chains detected could be on RIPK1-associated proteins, rather than RIPK1 itself.

Reply: To address this question, we repeated the experiments in Fig. 3a in denatured condition. We transfected HEK293T cells with myc-hRIPK1-WT or myc-hRIPK1-K627R with his-Ub or his-K48 only Ub or his-k63 only Ub. The cells were lysed in 8 M urea lysis buffer. The His-ubiquitinated RIPK1 was precipitated in denatured condition with Nickle-affinity resin, and the ubiquitination of RIPK1 was analyzed by western blotting using anti-myc antibody (Fig. 3a). The conclusion from new Fig. 3a suggests that the levels of K63 ubiquitinated RIPK1 species were predominantly reduced.

- Fig. 3d is presented as supporting evidence of K612 Ub being a master regulator of RIPK1 Ub, but this really is a stretch. All forms of TNF-induced Ub on RIPK1 are reduced because the recruitment of RIPK1 is impaired, and as discussed above, the authors have failed to prove that this recruitment defect stems from impaired Ub.

Reply: As described above, our new data on the ubiquitination levels of endogenous RIPK1 in WT and K612R knockin mutant MEFs showed that K612R mutation altered the levels of multiple ubiquitination sites on endogenous RIPK1 in unstimulated cells and also in 5 min TNF stimulated cells (Fig. 3f, Supplementary Fig. 5b). Thus, K612 is important for maintaining the levels of RIPK1 in control condition as well as in complex I. We do not claim that the recruitment defect is a direct consequence of impaired Ub.

- Fig. 4b: if males and females are used in this graph, then they should be presented separately because the average male body weight at 8 wks is greater than the average female body weight.

If a single sex is presented then this should be indicated in the legend.

Reply: To respond to this request, we provide the body weight data of male and female mice separately (Fig. 4b).

- Fig. 6d, h and text on page 16: absent any genetic rescue with MLKL loss, the authors cannot conclude that “necroptosis” is driving LPS-induced IL-1b release from KI BMDMs. RIPK3 has been shown to have non-necroptotic functions and thus its loss, while suggestive, is insufficient to prove necroptosis involvement.

Reply: We agree with this assessment. Our data showed that RIPK3 and RIPK3 kinase activity is indispensable in LPS-induced IL-1 β release in RIPK1-K612R BMDMs, but we have no evidence that MLKL may involved in this pathway. We have removed “necroptosis” from this sentence to say:

“Taken together, these results indicate that RIPK1^{K612R} mutation leads to disinhibition of RIPK3, which in turn promotes caspase-1 activation and IL-1 β cleavage upon LPS stimulation in BMDMs.”

Reviewer #2 (Remarks to the Author):

The manuscript by Li et al analyzed different ubiquitination sites in RIP1 for their function in cell death. K627 in human RIP1 was identified as a ubiquitination site that controls the overall ubiquitination of RIP1's kinase domain and DD domain mediated interaction with other proteins. K627R mutation inhibits TNF-induced cell death. The authors also generated RIP1 K612R mouse. In vitro and in vivo data showed that K612R mutation led to increased RIP3 dependent necroptosis, which sensitized cells to inflammasome activation by TLR3/4. The study was well designed and conducted. I have a few minor questions.

Reply: We thank the reviewer for strong support and helpful comments. We have carefully addressed each comment by editing and new experimental data.

1. Since caspase-1 activation and IL-1beta secretion occurred when K612 was mutated to R, a brief introduction and discussion of inflammasome is needed.

Reply: We have added a section in the Introduction and Discussion on inflammasome.

2. In K612R cells, complex I and II reduced, but necrosome increased. It is not very clear to me whether and how the impaired complex I formation by K612R mutation affected the formation of complex II and necrosome? Maybe a brief discussion is needed.

Reply: K612R functions differently in TNF and TLRs signaling pathway.

Dimerization of RIPK1 mediated by the DD is necessary for the RIPK1 kinase activation and complex II formation in TNFR1 signaling pathway. K612R not only inhibits DD-mediated RIPK1-RIPK1 dimerization, but also inhibits DD-mediated TRADD-RIPK1, FADD-RIPK1 and TNFR1-RIPK1 heterodimerization, which are all important for complex I and II formation in TNF signaling to promote necroptosis. In contrast, LPS stimulation of WT BMDMs cannot lead to RIPK1 activation, but can do so in K612R BMDMs (Fig. 6b,g). Thus, we believe that the ubiquitination of K612 normally suppresses the activation of RIPK1 in TLR pathway. This point is further clarified in the text and Discussion.

3. Page 13. The sentence "...lead to defective NF-kB pathway an a strong inflammatory response....." might cause confusion that defective NF-kB enhanced inflammation. It would be better to reword the sentence.

Reply: We have revised this sentence: "... leads to dysregulation of RIPK1 ubiquitination and a strong inflammatory response in the large intestine of adult *Ripk1*^{K612R/K612R} mice...".

4. Figure 6d. K62R/K612R needs to be corrected.

Reply: We thank this reviewer for the careful review. We corrected this typo.

5. The data in the manuscript seem to suggest that K612R did not simply mimic un-ubiquitinated RIP1, but also gained additional function(s). The authors need to discuss this issue

Reply: This reviewer is absolutely correct that K612R does not simply mimic un-ubiquitinated RIPK1. As shown in the new Fig. 3f, supplementary Fig.5b, we have further characterized the

sites and levels of dynamic ubiquitination on endogenous RIPK1 in WT and K612R knockin MEFs under control condition and after stimulation by TNF α for 5 min. We immunoprecipitated endogenous RIPK1 from RIPK1-WT-MEFs and RIPK1-K612R knockin MEFs that were treated with or without TNF α for 5 mins using a validated rabbit monoclonal anti-RIPK1 antibody and analyzed the ubiquitination patterns of WT and K612R RIPK1 by quantitative mass spectrometry. In WT MEFs under control condition without TNF α stimulation, many lysine residues in RIPK1 were already highly ubiquitinated, including 11 lysine residues in the kinase domain (K20, K30, K45, K65, K115, K137, K140, K153, K163, K167 and K307), 4 lysine residues in the intermediate domain (K376, K392, K395, and K429) and 4 lysine residues in the DD (K589, K612, K627 and K633) (Supplementary Fig. 5b). Similar to the mass spec data on the ubiquitination changes of K612R RIPK1 expressed in 293T cells, the majority of these sites in the kinase domain and intermediate domain of endogenous K612R RIPK1 showed a reduction in ubiquitination levels under control condition (Fig. 3e, f; Supplementary Fig. 5b). The ubiquitination levels of 15 lysine residues (out of the 19 ubiquitinated lysine residues in WT-RIPK1) on K612R-RIPK1 in K612R-RIPK1 knockin MEFs under control condition were reduced compared to that of WT RIPK1 in WT MEFs, including K20, K30, K45, K46, K115, K137, K140, K153, K163, K167, K307, K392, K395, K429 and K627 (Fig. 3f and Supplementary Fig. 5b). The ubiquitination levels of K65 and K376 on K612R-RIPK1 in RIPK1-K612R knockin MEFs under control conditions were comparable to that of WT-RIPK1 in WT MEFs; while 3 lysine residues in the DD of K612R-RIPK1 (K589, K619 and K633) and one lysine residues in the kinase domain (K105) in K612R-RIPK1 knockin MEFs were increased compared to that of WT-RIPK1. With the majority of ubiquitination events in the kinase domain and the intermediate domain reduced while the ubiquitination of the DD increased by K612R mutation, the changes in overall ubiquitination patterns on endogenous RIPK1 by K612R mutation were similar to that of K627R RIPK1 expressed in 293T cells (Fig. 3e, f).

We further used quantitative mass spectrometry to compare the patterns of ubiquitination in the endogenous WT RIPK1 and K612R RIPK1 in WT and RIPK1-K612R knockin MEFs stimulated by TNF α for 5 min. In WT MEFs stimulated by TNF α for 5 min, the ubiquitination levels of 19 lysine residues, including K20, K45, K46, K65, K105, K137, K140, K153, K163, K167, K376, K392, K395, K429, K550, K589, K612, K619 and K633, were increased; the ubiquitination levels of 4 lysine residues, including K30, K115, K307 and K627, were decreased (Fig 3f; Supplementary Fig. 5b). In K612R-RIPK1 knockin MEFs stimulated by TNF α for 5 min, the ubiquitination levels of 17 lysine residues in K612R-RIPK1 were reduced compared to that of RIPK1 in WT MEFs, including K20, K30, K45, K46, K65, K105, K137, K140, K153, K163, K167, K376, K392, K395, K429, K550, K619; while the ubiquitination levels of K589 were comparable to that of WT; and the ubiquitination levels of 3 lysine residues in K612R-RIPK1 in K612R-RIPK1 knockin MEFs, including K115, K307 and K633, were increased compared to that of WT (Fig. 3f; Supplementary Fig. 5b). Thus, while TNF α stimulation led to a rapid increase in the ubiquitination of multiple sites over entire RIPK1, the ubiquitination levels of many of the lysine residues in K612R RIPK1 were blocked or reduced (Fig 3f; Supplementary Fig. 5b). Since K612R mutation affects the levels and patterns of RIPK1 ubiquitination under both control condition and after stimulation by TNF α for 5 min, our results suggest that the ubiquitination of K612 can affect the overall pattern of

RIPK1 ubiquitination in the cytoplasm under unstimulated condition as well as after TNF α stimulation which can affect its recruitment to complex I.

Reviewer #3 (Remarks to the Author):

In this manuscript, Li et al. present convincing evidence that K612 in mRIPK1/K627 in hRIPK1 plays a critical role in regulating TNFR1 and TLR signaling as a master ubiquitination site that controls the ubiquitination pattern of RIPK1. They generated Ripk1K612R/K612R knockin mutant mice to extensively dissect the function of K612 in vitro and in vivo. Interestingly, they provided evidence that K612R mutation inhibits TNF α induced-RDA and necroptosis, but promotes necroptosis and caspase-1 activation stimulated by TLR3/4. Ripk1K612R/K612R mice develop adult-onset intestinal inflammation and splenomegaly, which can be rescued by antibiotic treatment or co-ablation of Ripk3. Further investigation showed that RIPK1 K612 is important for DD mediated interaction of RIPK1 with different DD-containing proteins, which may explained the distinct function of K612 in TNFR1 and TLRs signaling. The data shown are of very good quality. Here are several aspects that need improvement.

Reply: We thank the reviewer for the strong support and helpful suggestions. We have carefully addressed each comment by editing and new experimental data.

1. In Figure 3b, M1 ubiquitination of hRIPK1 K627R is shown to be comparable with that of WT, but M1 and K63 ubiquitinated RIPK1 in K612R mutations cells were reduced compared to that of WT cells upon TNF α /SM-164/zVAD treatment. This difference should be discussed.

Reply: To address this question, we used ImageJ to quantitatively measure the difference in M1 and K63 ubiquitination in overexpressed WT and K627R-RIPK1 knockin MEFs. Our data suggest that in both systems, both M1 and K63 ubiquitinations of RIPK1 were reduced (Fig. 3c).

2. In Figure 4, authors only provided results of macrophages and neutrophil in colon. Other type immune cells should also be examined in thymus, spleen and lymph nodes to provide a complete picture of immune homeostasis in Ripk1K612R/K612R mice. Was there accompanied with lymphadenopathy in Ripk1K612R/K612R mice?

Reply: Ripk1^{K612R/K612R} mice displayed progressive spleen enlargement, while the size of thymus and lymph nodes were normal as that of WT mice until 20 weeks of age (oldest mice observed). Thus, unlike the spleens of Fadd^{-/-}Mkl1^{-/-} and Fadd^{-/-}Ripk3^{-/-} mice which accumulate an abnormal large population of B220⁺CD3⁺ T lymphocytes^{4,5}, B220⁺CD3⁺ T lymphocytes were not significantly increased in spleens of Ripk1^{K612R/K612R} mice. Thus, Ripk1^{K612R/K612R} mice do not develop the lymphoproliferative disease as that of Fadd^{-/-}Mkl1^{-/-} and Fadd^{-/-}Ripk3^{-/-} mice (Supplementary Fig. 6d-f).

3. In Figure 4c, RIPK1 protein level in multiple tissue from aged Ripk1K612R/K612R mice was greatly decreased compared to that from WT mice. Authors should examine if RIPK1 K612R has moved into the insoluble fraction.

Reply: To address this question, we examined RIPK1 protein level in soluble and insoluble fraction. Our results showed that RIPK1 protein levels in both soluble and insoluble fraction were decreased in the colon and ileum tissues of at 20 weeks-old Ripk1^{K612R/K612R} mice (Supplementary Fig. 6a). These results suggest that although no difference in the levels of

RIPK1 was detected in tissues of newborns, there is a progressive reduction in the levels of RIPK1 in K612R mice.

4. In addition, whether K612 in mRIPK1/K627 in hRIPK1 controlling the ubiquitination pattern on other sites of RIPK1 is mediated by RIPK1 kinase activity should be test or at least discussed in text.

Reply: To address this question, we further investigated whether K612R mutation could affect the ubiquitination of endogenous RIPK1 by quantitative mass spectrometry analysis. We immunoprecipitated endogenous RIPK1 from RIPK1-WT-MEFs and RIPK1-K612R knockin MEFs that were treated with or without TNF α for 5 mins using a validated rabbit monoclonal anti-RIPK1 antibody and analyzed the ubiquitination patterns of WT and K612R RIPK1 by quantitative mass spectrometry. In WT MEFs under control condition without TNF α stimulation, many lysine residues in RIPK1 were already highly ubiquitinated, including 11 lysine residues in the kinase domain (K20, K30, K45, K65, K115, K137, K140, K153, K163, K167 and K307), 4 lysine residues in the intermediate domain (K376, K392, K395, and K429) and 4 lysine residues in the DD (K589, K612, K627 and K633) (Supplementary Fig. 5b). Similar to the mass spec data on the ubiquitination changes of K612R RIPK1 expressed in 293T cells, the majority of these sites in the kinase domain and intermediate domain of endogenous K612R RIPK1 showed a reduction in ubiquitination levels under control condition (Fig. 3e, f; Supplementary Fig. 5b). The ubiquitination levels of 15 lysine residues (out of the 19 ubiquitinated lysine residues in WT-RIPK1) on K612R-RIPK1 in K612R-RIPK1 knockin MEFs under control condition were reduced compared to that of WT RIPK1 in WT MEFs, including K20, K30, K45, K46, K115, K137, K140, K153, K163, K167, K307, K392, K395, K429 and K627 (Fig. 3f and Supplementary Fig. 5b). The ubiquitination levels of K65 and K376 on K612R-RIPK1 in RIPK1-K612R knockin MEFs under control conditions were comparable to that of WT-RIPK1 in WT MEFs; while 3 lysine residues in the DD of K612R-RIPK1 (K589, K619 and K633) and one lysine residues in the kinase domain (K105) in K612R-RIPK1 knockin MEFs were increased compared to that of WT-RIPK1. With the majority of ubiquitination events in the kinase domain and the intermediate domain reduced while the ubiquitination of the DD increased by K612R mutation, the changes in overall ubiquitination patterns on endogenous RIPK1 by K612R mutation were similar to that of K627R RIPK1 expressed in 293T cells (Fig. 3e, f).

We further used quantitative mass spectrometry to compare the patterns of ubiquitination in the endogenous WT RIPK1 and K612R RIPK1 in WT and RIPK1-K612R knockin MEFs stimulated by TNF α for 5 min. In WT MEFs stimulated by TNF α for 5 min, the ubiquitination levels of 19 lysine residues, including K20, K45, K46, K65, K105, K137, K140, K153, K163, K167, K376, K392, K395, K429, K550, K589, K612, K619 and K633, were increased; the ubiquitination levels of 4 lysine residues, including K30, K115, K307 and K627, were decreased (Fig 3f; Supplementary Fig. 5b). In K612R-RIPK1 knockin MEFs stimulated by TNF α for 5 min, the ubiquitination levels of 17 lysine residues in K612R-RIPK1 were reduced compared to that of RIPK1 in WT MEFs, including K20, K30, K45, K46, K65, K105, K137, K140, K153, K163, K167, K376, K392, K395, K429, K550, K619; while the ubiquitination levels of K589 were comparable to that of WT; and the ubiquitination levels of 3 lysine residues in K612R-RIPK1 in K612R-RIPK1 knockin MEFs, including K115, K307

and K633, were increased compared to that of WT (Fig. 3f; Supplementary Fig. 5b). Thus, while TNF α stimulation led to a rapid increase in the ubiquitination of multiple sites over entire RIPK1, the ubiquitination levels of many of the lysine residues in K612R RIPK1 were blocked or reduced (Fig 3f; Supplementary Fig. 5b). Since K612R mutation affects the levels and patterns of RIPK1 ubiquitination under both control condition and after stimulation by TNF α for 5 min, our results suggest that the ubiquitination of K612 can affect the overall pattern of RIPK1 ubiquitination in the cytoplasm under unstimulated condition as well as after TNF α stimulation which can affect its recruitment to complex I.

Reference:

1. Dillon CP, *et al.* RIPK1 blocks early postnatal lethality mediated by caspase-8 and RIPK3. *Cell* **157**, 1189-1202 (2014).
2. Conos SA, *et al.* Active MLKL triggers the NLRP3 inflammasome in a cell-intrinsic manner. *Proc Natl Acad Sci U S A* **114**, E961-E969 (2017).
3. Meng H, *et al.* Death-domain dimerization-mediated activation of RIPK1 controls necroptosis and RIPK1-dependent apoptosis. *Proc Natl Acad Sci U S A* **115**, E2001-E2009 (2018).
4. Zhang X, *et al.* MLKL and FADD Are Critical for Suppressing Progressive Lymphoproliferative Disease and Activating the NLRP3 Inflammasome. *Cell Rep* **16**, 3247-3259 (2016).
5. Alvarez-Diaz S, *et al.* The Pseudokinase MLKL and the Kinase RIPK3 Have Distinct Roles in Autoimmune Disease Caused by Loss of Death-Receptor-Induced Apoptosis. *Immunity* **45**, 513-526 (2016).

REVIEWERS' COMMENTS

Reviewer #1 (Remarks to the Author):

I think the layout of the revision is much improved and I believe the data. I am just not 100% convinced that the phenotypes described for the K612R mutant are a direct consequence of impaired ubiquitination at this residue. It seems that they could equally well stem from an improperly folded death domain (DD) and compromised DD-DD protein interactions. Thus, the main focus of the paper "functions of RIPK1 ubiquitination" is in doubt. Instead, they can only conclude that residue K612 within the DD is important for normal RIPK1 function, which isn't particularly enlightening. Unfortunately, not knowing the ubiquitin ligase that modifies RIPK1 on K612, they can't validate their model by testing if ligase deficiency gives the same result as the K612R mutation.

Reviewer #2 (Remarks to the Author):

The authors addressed my questions. I have no further comment.

Reviewer #3 (Remarks to the Author):

This revised manuscript by LI et al. has substantially improved by modifying the manuscript text and by the additional of several important experiments presented in both main and supplemental figures. The results presented in this study advance our understanding of the molecular mechanisms that regulate the RIPK1 activation by its ubiquitination in response to TNFR1 and TLR signaling.

REVIEWERS' COMMENTS

Reviewer #1 (Remarks to the Author):

I think the layout of the revision is much improved and I believe the data. I am just not 100% convinced that the phenotypes described for the K612R mutant are a direct consequence of impaired ubiquitination at this residue. It seems that they could equally well stem from an improperly folded death domain (DD) and compromised DD-DD protein interactions. Thus, the main focus of the paper "functions of RIPK1 ubiquitination" is in doubt. Instead, they can only conclude that residue K612 within the DD is important for normal RIPK1 function, which isn't particularly enlightening. Unfortunately, not knowing the ubiquitin ligase that modifies RIPK1 on K612, they can't validate their model by testing if ligase deficiency gives the same result as the K612R mutation.

Reply: We thank this reviewer for the comments and support. Based on our study, human RIPK1 K627R and murine RIPK1 K612R mutations can both dramatically affect the overall ubiquitination pattern of RIPK1. Because Lysine and Arginine are very similar in structure, polarity and charge, it is unlikely that a change from Lys to Arg would make a protein to misfold. Given the critical role of ubiquitination in regulating the activation of RIPK1, it is most likely that the changes in the ubiquitination pattern of K612R plays a major role in directing the activation of this RIPK1 mutant. Given this is already a 7-year project by itself, finding the E3 ligase for K612/K627 can be a project for next graduate student/postdoc.

Reviewer #2 (Remarks to the Author):

The authors addressed my questions. I have no further comment.

Reply: We thank very much for the reviewer's support for this revised manuscript.

Reviewer #3 (Remarks to the Author):

This revised manuscript by LI et al. has substantially improved by modifying the manuscript text and by the additional of several important experiments presented in both main and supplemental figures. The results presented in this study advance our understanding of the molecular mechanisms that regulate the RIPK1 activation by its ubiquitination in response to TNFR1 and TLR signaling.

Reply: We thank very much for the reviewer's support for this revised manuscript.